# Geometric Convergence of Gauss–Newton for Neural Networks: Riemannian Geometry and Adaptive Damping

**Semih Cayci** [1]

## Abstract

Ill-conditioned kernel matrices can make first-order methods for training neural networks converge slowly. We establish non-asymptotic convergence bounds for the Gauss–Newton method in both under- and overparameterized regimes, showing it avoids these conditioning bottlenecks. In the underparameterized setting, Gauss–Newton gradient flow in parameter space induces a Riemannian gradient flow on a low-dimensional submanifold of function space. Using tools from Riemannian optimization, we show that, under an appropriate output scaling, the loss satisfies geodesic Polyak–Lojasiewicz and Lipschitz-smoothness conditions, implying geometric convergence to the optimal in-class predictor at an explicit rate independent of Gram-matrix conditioning. In the overparameterized setting, we identify adaptive, curvature-aware regularization schedules and prove fast geometric convergence to a global optimum for both Gauss–Newton gradient flow and discrete-time Gauss–Newton iterates, with rates independent of the minimum eigenvalue of the neural tangent kernel. Overall, Gauss–Newton can be provably faster in ill-conditioned regimes where first-order methods slow down.

## 1. Introduction

First-order optimization methods can converge slowly in neural network training, often due to ill-conditioned data geometry and loss landscapes (Shalev-Shwartz et al., 2017). This has motivated geometry-aware, preconditioned methods that aim to stabilize training and improve convergence. Among them, the Gauss–Newton (GN) method has drawn sustained practical and theoretical interest, driven by strong empirical performance at scale (Abreu et al., 2026; Liu et al., 2025; Martens, 2020; Ren & Goldfarb, 2019; Botev et al., 2017). Yet, the theoretical understanding of GN in deep learning remains limited, particularly regarding nonasymptotic convergence and the role of preconditioning in under- and overparameterized regimes.

### 1.1. Main Contributions

In this work, we investigate the convergence behavior of the Gauss–Newton method for training neural networks, where we establish a Riemannian geometric analysis in the underparameterized regime and a curvature-aware damping analysis in the overparameterized kernel regime.

- **Overparameterized regime (adaptive damping).** We identify curvature-aware damping schedules for Gauss–Newton that yield *conditioning-robust* geometric convergence to a global optimum, for both the Gauss–Newton gradient flow and discrete-time Gauss–Newton iterates. In particular, the convergence rates of Gauss–Newton are independent of the minimum eigenvalue of the kernel matrix. Consequently, the convergence rates of the Gauss–Newton method are robust to ill-conditioned kernel matrices that can slow down standard first-order methods. (See Theorems 3.4, 3.8 and Corollary. 3.6; extensions to deep networks are in Section 3.2.)

- **Underparameterized regime (Riemannian optimization).** We show that, in the underparameterized setting, the Gauss–Newton gradient flow in parameter space induces a Riemannian gradient flow on a low-dimensional embedded submanifold of the predictor space. Using tools from Riemannian optimization, we establish geodesic Polyak–Lojasiewicz and geodesic smoothness of the loss on a level set containing the optimization trajectory under an appropriate output scaling, which explicitly controls curvature. These regularity properties imply *last-iterate* geometric convergence to the optimal in-class predictor at an explicit rate independent of Gram-matrix conditioning, without any explicit regularization. (See Theorems 4.10 and 4.14, and Propositions 4.6 and 4.8.)

[1]Department of Mathematics, RWTH Aachen University, Aachen, Germany. Correspondence to: Semih Cayci <cayci@mathc.rwth-aachen.de>.

*Proceedings of the 43rd International Conference on Machine Learning*, Seoul, South Korea. PMLR 306, 2026. Copyright 2026 by the author(s).

## 1.2. Related Works

**Analysis of the Gauss–Newton method.** The Gauss–Newton method has a long history in numerical linear algebra (Saad, 2003) and nonlinear least-squares (Nocedal & Wright, 1999). It has recently attracted renewed attention because of its success in deep learning (Abreu et al., 2026; Liu et al., 2025; Korbit et al., 2025; Tan & Lim, 2019; Botev et al., 2017) and scientific machine learning (Rathore et al., 2024; Hao et al., 2024). Despite its empirical success, its theoretical understanding in modern deep learning remains limited. Its convergence has been investigated in a number of recent works (Cai et al., 2019b; Zhang et al., 2019; Arbel et al., 2024; Adeoye et al., 2024; Zhao et al., 2024; Jia et al., 2024), which study Gauss–Newton in overparameterized regimes. These prior works largely focus on quadratic loss settings (Cai et al., 2019b; Zhang et al., 2019) or mean-field limits (Arbel et al., 2024). In this work, we show that curvature-aware damping yields geometric convergence bounds whose rate does not deteriorate with $\lambda_{\min}(K_0)$. We also provide finite-time and finite-width guarantees in the underparameterized regime via a novel Riemannian analysis of Gauss–Newton, which is fundamentally different from existing approaches.

**Optimization in the lazy training regime.** The original works in the lazy training regime analyze the convergence of gradient descent for overparameterized neural networks (Du et al., 2018; Jacot et al., 2018; Chizat et al., 2019). Our analysis builds on the near-initialization analysis proposed in (Chizat et al., 2019; Du et al., 2018), and extends it to analyze the Gauss–Newton method for both over- and underparameterized neural networks. In the underparameterized regime, deviating significantly from the existing works, we integrate tools from the Riemannian optimization theory to analyze the Gauss–Newton dynamics in training neural networks. In a number of works (Ji & Telgarsky, 2020; Cayci & Eryilmaz, 2025; Cai et al., 2019a), convergence of first-order methods in the underparameterized regime was investigated in the near-initialization regime. These results establish near-optimality results for first-order methods (i) under explicit regularization in the form of projection or early stopping, (ii) for the average- or best-iterate, and (iii) with convergence rates into a ball around the near-optimal parameter at a slow subexponential rate. Chizat et al. (2019) establishes the convergence of gradient flow in the underparameterized regime, but the rate is unspecified and has a strong dependence on the spectrum of the Gramian at initialization. On the other hand, we prove that Gauss–Newton dynamics (i) achieves last-iterate *convergence* to an in-class optimum predictor, (ii) without any realizability assumptions, and (iii) at a fast geometric convergence rate independent of the Gramian at initialization, emphasizing the benefits of GN in the underparameterized regime.

## 1.3. Notation

For a differentiable curve $\gamma : I \subset \mathbb{R}^+ \to \mathbb{R}$, $\dot{\gamma}_t$ and $\gamma'(t)$ denote its derivative at time $t$. $\mathrm{Lip}_f$ denotes the modulus of Lipschitz continuity of $f$. For a symmetric positive-definite $\boldsymbol{A} \in \mathbb{R}^{n \times n}$ and $v \in \mathbb{R}^n$, $\|x\|_{\boldsymbol{A}}^2 := x^\top \boldsymbol{A} x$. For $\boldsymbol{P} \in \mathbb{R}^{n \times n}$, $\|\boldsymbol{P}\|$ denotes its operator norm and $\lambda_{\min}(\boldsymbol{P})$ denotes its minimum eigenvalue.

**Conflict of Interest Disclosure.** The author declares no financial conflicts of interest.

# 2. Problem Setting and the Gauss–Newton Dynamics

## 2.1. Supervised Learning Setting

In this work, we consider a supervised learning problem with a data set $\mathcal{D} = \{(x_j, y_j)\}_{j=1}^n$, $x_j \in \mathbb{R}^d, y_j \in \mathbb{R}$. Given a loss function $\ell : \mathbb{R} \times \mathbb{R} \to \mathbb{R}_+$, the empirical risk under the prediction $\xi \in \mathbb{R}^n$ is $g(\xi) := \sum_{j=1}^n \ell(\xi_j, y_j)$. We define $X := \max_{j \in [n]} \|x_j\|_2^2$.

**Deep fully-connected neural networks.** We consider a feedforward deep neural network of depth $H \geq 1$ and width $m$ with a smooth (infinitely-differentiable) activation function $\sigma : \mathbb{R} \to \mathbb{R}$. We assume $\|\sigma'\|_\infty \leq \sigma_1 < \infty$ and $\|\sigma''\|_\infty \leq \sigma_2 < \infty$, which holds for activations such as $\tanh$, sigmoid and GELU. Let $W^{(1)} \in \mathbb{R}^{m \times d}$ and $W^{(h)} \in \mathbb{R}^{m \times m}$ for $h = 2, 3, \ldots, H$, and $\boldsymbol{W} := (W^{(1)}, \ldots, W^{(H)})$. Then, given a training input $x_j \in \mathbb{R}^d$, the neural network is defined recursively as

$$\boldsymbol{x}_j^{(h)}(\boldsymbol{W}) = \sqrt{\frac{a_\sigma}{m}} \cdot \vec{\sigma}\left(W^{(h)} \boldsymbol{x}_j^{(h-1)}(\boldsymbol{W})\right), \quad h \in [H],$$

$$\varphi(x_j; w) = c^\top \boldsymbol{x}_j^{(H)}(\boldsymbol{W}),$$

where $\boldsymbol{x}_j^{(0)}(\boldsymbol{W}) = x_j$, $a_\sigma := (\mathbb{E}_{z \sim \mathcal{N}(0,1)}[\sigma^2(z)])^{-1}$ is normalization parameter, $w = \mathrm{vec}(\boldsymbol{W}, c)$ is the parameter vector, and $\vec{\sigma}(z) = [\sigma(z_1) \ \ldots \ \sigma(z_m)]^\top$.

**Random initialization.** We adopt the standard NTK initialization $w_0 = (c_0, \boldsymbol{W}_0)$ as in (Ji & Telgarsky, 2020; Du et al., 2018): for each layer $h \in [H]$,

$$[c_0]_i \overset{\mathrm{iid}}{\sim} \mathrm{Rad} \quad \text{and} \quad [W_0^{(h)}]_{ij} \overset{\mathrm{iid}}{\sim} \mathcal{N}(0, 1). \tag{1}$$

We denote the prediction function as

$$f(w) := [\varphi(x_1; w) \ \ldots \ \varphi(x_n; w)] - b,$$

where $b \in \mathbb{R}^n$ is a fixed bias term. To ensure $f(w_0) = 0$, as in (Chizat et al., 2019), we set $b_j = \varphi(x_j; w_0)$, $j \in [n]$, which simply re-centers the model in function space. Since $b$ is constant, all derivatives with respect to $w$ are unchanged.

**Optimization in supervised learning.** The objective in this paper is empirical risk minimization:

$$\min_{w \in \mathbb{R}^p} \ g(\alpha f(w)), \tag{2}$$

where $\alpha > 0$ is an output scaling parameter, which plays a key role in our convergence results. For convenience, we define

$$\mathcal{R}(w) := g(\alpha f(w)), \quad w \in \mathbb{R}^p.$$

We assume $g$ is $\nu$-strongly convex and has $\mu$-Lipschitz continuous gradients as a function of the prediction $\xi \in \mathbb{R}^n$:

$$\nu \boldsymbol{I} \preccurlyeq \nabla^2 g(\xi) \preccurlyeq \mu \boldsymbol{I}, \quad \xi \in \mathbb{R}^n. \tag{3}$$

For quadratic loss $g(\xi) = \frac{1}{2} \sum_{j=1}^{n} (\xi_j - y_j)^2$, we have $\mu = \nu = 1$. Note that although $\xi \mapsto g(\xi)$ is strongly convex in the prediction $\xi \in \mathbb{R}^n$, $w \mapsto g(\alpha f(w)) =: \mathcal{R}(w)$ is nonconvex as a function of the neural network parameters $w \in \mathbb{R}^p$, which is the main challenge in the empirical risk minimization problem with neural networks. The number of learnable parameters is $p$, i.e., $w \in \mathbb{R}^p$. We call the model *underparameterized* if $p \leq n$, and *overparameterized* otherwise. For each layer $h \in [H]$, we denote the Jacobians of $f(w)$ with respect to $W^{(h)}$ and $w$, respectively, as

$$[\mathrm{D}_h f(w)]_{j,:} := \nabla_{\mathrm{vec}(W^{(h)})}^\top \varphi(x_j; w)$$
$$[\mathrm{D} f(w)]_{j,:} := \nabla_w^\top \varphi(x_j; w), \ j \in [n].$$

### 2.2. Gauss–Newton Gradient Flow

We consider the following preconditioned gradient flow to solve (2):

$$\dot{w}_t = -\frac{1}{\alpha} \big[ \boldsymbol{H}(w_t) \big]^{-1} \mathrm{D}^\top f(w_t) \nabla g(\alpha f(w_t)), \tag{4}$$

where $\alpha > 0$ is the output scaling parameter, and $\boldsymbol{H}(w) \in \mathbb{R}^{p \times p}$ is the preconditioner. The *generalized Gauss–Newton* (GGN) preconditioner is $\boldsymbol{H} = \boldsymbol{H}_\rho$, where

$$\boldsymbol{H}_\rho(w) := (1 - \rho(w))\boldsymbol{\Gamma}(w) + \rho(w)\boldsymbol{I} \tag{5}$$
$$\boldsymbol{\Gamma}(w) := \mathrm{D}^\top f(w) \nabla^2 g(\alpha f(w)) \mathrm{D} f(w)$$

with the damping (Levenberg–Marquardt regularization) factor $\rho : \mathbb{R}^p \to [0, 1]$. We also study the (classical) Gauss–Newton flow with $\boldsymbol{H} = \boldsymbol{H}_\rho^{\mathrm{G}}$, where

$$\boldsymbol{H}_\rho^{\mathrm{G}}(w) := (1 - \rho(w))\boldsymbol{\Gamma}^{\mathrm{G}}(w) + \rho(w)\boldsymbol{I}, \tag{6}$$
$$\boldsymbol{\Gamma}^{\mathrm{G}}(w) := \mathrm{D}^\top f(w) \mathrm{D} f(w).$$

The GGN preconditioner $\boldsymbol{H}_\rho$ reduces to $\boldsymbol{H}_\rho^{\mathrm{G}}$ for quadratic loss $g(\xi) = \frac{1}{2} \sum_{j=1}^{n} (\xi_j - y_j)^2$, since $\nabla^2 g \equiv \boldsymbol{I}$. We analyze (4) with $\boldsymbol{H} = \boldsymbol{H}_\rho^{\mathrm{G}}$ for general (possibly non-quadratic) $g$ satisfying (3) (cf. (Zhao et al., 2024)), and then quantify the additional benefits of using $\boldsymbol{H}_\rho$ when $\nabla^2 g(\alpha f(w_t))$ varies along the optimization trajectory.

## 3. Adaptive Damping and Convergence in the Overparameterized Regime

We begin with the overparameterized regime $p > n$. Since $\mathrm{rank}(\mathrm{D} f(w)) \leq n < p$ in this regime, the Gauss–Newton matrix is rank-deficient, therefore we study the damped Gauss–Newton flow with $\rho > 0$, i.e., Levenberg–Marquardt dynamics. The choice of damping schedule $\rho(\cdot)$ is important (Arbel et al., 2024). In this work, we identify damping schedules that yield provably fast convergence rates that do not depend on the minimum eigenvalue of the kernel matrix.

We first present the analysis in the case $H = 1$ with the output layer $c$ frozen at initialization, which isolates the main ideas with sharp bounds and minimal notation (Chizat et al., 2019; Du et al., 2018). In this setting, $w = \mathrm{vec}(W^{(1)})$ and $p = md$. We then extend the arguments to deep networks with trainable $c$ in Corollary 3.11.

### 3.1. Gauss–Newton for Single-Hidden Layer Networks

In the overparameterized regime, the spectral properties of the kernel matrix have a crucial impact on the convergence. To that end, define $\boldsymbol{K}, \bar{\boldsymbol{K}} \in \mathbb{R}^{n \times n}$ by

$$[\bar{\boldsymbol{K}}]_{ij} := x_i^\top x_j \mathbb{E}_{u_0 \sim \mathcal{N}(0, \boldsymbol{I}_d)}[\sigma'(u_0^\top x_i)\sigma'(u_0^\top x_j)]a_\sigma^2,$$
$$\boldsymbol{K}(w) := \mathrm{D} f(w) \mathrm{D}^\top f(w), \quad w \in \mathbb{R}^p.$$

Note that under the initialization $(c_0, w_0)$, we have $\mathbb{E}[\boldsymbol{K}(w_0)] = \bar{\boldsymbol{K}}$. We make the following standard representational assumption on the neural tangent kernel matrix at initialization (Chizat et al., 2019).

**Assumption 3.1.** Assume that $\boldsymbol{K}(w_0)$ is strictly positive definite with the minimum eigenvalue $4\lambda^2 > 0$.

*Remark* 3.2 ($\lambda_{\min}(\boldsymbol{K}(w_0))$ and data geometry). The geometry of $\{x_i \in \mathbb{R}^d\}_{i=1}^n$ strongly influences $\lambda_{\min}(\boldsymbol{K}(w_0))$. If no input pairs are parallel, then Assumption 3.1 automatically holds (Du et al., 2018). $\lambda$ can be upper bounded in terms of $\delta'(\mathcal{D}) := \min_{i \neq j} \|x_i - x_j\|_2$ (Karhadkar et al., 2024). Thus, $\boldsymbol{K}(w_0)$ can be highly ill-conditioned (with extremely small $\lambda^2$) in practice, which motivates condition-robust convergence rates.

For a single-hidden layer network, $w \mapsto f(w)$ has globally $L$-Lipschitz gradients with $L = \sigma_2 X \sqrt{\frac{n}{m}}$. If

$$\|w - w_0\|_2 < r_0 := \lambda/L, \tag{7}$$

then $\mathrm{D} f(w) \mathrm{D}^\top f(w) \succcurlyeq \lambda^2 \boldsymbol{I}$ (Chizat et al., 2019). Define

$$T := \inf\{t > 0 : \|w_t - w_0\|_2 \geq r_0\}$$

as the first-exit time. Also, let $\boldsymbol{K}_t := \boldsymbol{K}(w_t)$, $t \in [0, \infty)$ be the kernel matrix, and $\lambda_t^2 := \lambda_{\min}(\boldsymbol{K}_t)$. Then, we have

$$\inf_{t \in [0,T)} \lambda_t^2 \geq \lambda^2. \tag{8}$$

### 3.1.1. GAUSS–NEWTON: CONDITIONING-ROBUST GLOBAL CONVERGENCE RATES

We first analyze the convergence of (4) with the classical Gauss–Newton preconditioner $\boldsymbol{H} = \boldsymbol{H}_\rho^{\mathrm{G}}$ for any loss $g$ satisfying (3).

The evolution in function space under any damping scheme $\{\rho_t \in (0,1]\}_{t \geq 0}$ are presented in the following lemma.

**Lemma 3.3.** *Under the GN flow with any* $\{\rho_t \in (0,1]\}_{t \geq 0}$,

$$\frac{\mathrm{d}\alpha f(w_t)}{\mathrm{d}t} = -\boldsymbol{\Sigma}_t(\boldsymbol{K}_t)\nabla g(\alpha f(w_t)), \qquad \text{(GF-O)}$$

$$\frac{\mathrm{d}g(\alpha f(w_t))}{\mathrm{d}t} \leq \frac{-\lambda_t^2}{\rho_t + (1-\rho_t)\lambda_t^2}\|\nabla g(\alpha f(w_t))\|_2^2, \quad \text{(EDI)}$$

*where* $\boldsymbol{\Sigma}_t(\boldsymbol{K}) := \frac{1}{\rho_t}\left(\boldsymbol{K} - \frac{1-\rho_t}{\rho_t}\boldsymbol{K}\left(\boldsymbol{I} + \frac{1-\rho_t}{\rho_t}\boldsymbol{K}\right)^{-1}\boldsymbol{K}\right)$ *for any* $t < T$.

Consider the adaptive damping schedule

$$\rho^{\mathrm{ada}}(w) = \frac{\lambda_{\min}(\boldsymbol{K}(w))}{1 + \lambda_{\min}(\boldsymbol{K}(w))}. \tag{9}$$

GN flow under this damping factor yields geometric convergence at a fast rate that does not deteriorate as $\lambda_{\min}(\boldsymbol{K}_0) \downarrow 0$. Intuitively, this choice gives more weight to GN when $\lambda_{\min}(\boldsymbol{K}(w))$ is small and GF is slow (see (GF-O)).

**Theorem 3.4** (Convergence of GN – adaptive damping). *Under Assumption 3.1, the Gauss–Newton flow (4) with* $\boldsymbol{H} = \boldsymbol{H}_\rho^{\mathrm{G}}$ *and the adaptive damping* $\rho_t = \rho^{\mathrm{ada}}(w_t)$, $t \geq 0$ *yields* $T = \infty$ *(i.e., admits a global solution) for any output scaling parameter* $\alpha > 0$ *such that*

$$\alpha\sqrt{m} \geq \frac{\sigma_2\mu X\sqrt{2nV_0}}{\nu^{3/2}} \cdot \frac{1 + \lambda\mathrm{Lip}_f}{\lambda^2(1+\lambda^2)}.$$

*Consequently, the following bounds hold for all* $t \geq 0$:

$$V_t \leq V_0 \cdot \exp\left(-\nu(1+\lambda^2)t\right),$$

$$\|\alpha f(w_t) - f^\star\|_2^2 \leq \frac{\mu}{\nu} \cdot \|f^\star\|_2^2 \cdot \exp\left(-\nu(1+\lambda^2)t\right), \tag{10}$$

*where* $f^\star \in \arg\min_{\xi \in \mathbb{R}^n} g(\xi)$ *is the unique minimizer of* $g$ *and*

$$V_t := g(\alpha f(w_t)) - g(f^\star), \quad t \geq 0$$

*is the optimality gap.*

The proof of Theorem 3.4 is given in Appendix A.

*Remark 3.5.* We have the following observations:

- GN with adaptive damping achieves

$$V_t \leq V_0 e^{-\nu(1+\lambda^2)t} \leq V_0 e^{-\nu t}, \qquad t \geq 0,$$

so the decay rate is uniformly bounded below by $\nu > 0$, and does not deteriorate as $\lambda^2 = \lambda_{\min}(\boldsymbol{K}_0)/4 \downarrow 0$. In contrast, gradient flow has a decay rate $(\nu/2)\lambda_{\min}(\boldsymbol{K}_0)$ in the same setting (Chizat et al., 2019), which can be arbitrarily slow when $\lambda_{\min}(\boldsymbol{K}_0) \downarrow 0$. Note that GN has a polynomial dependence of $\alpha\sqrt{m}$ on $\lambda^{-1}$ similar to GF (Chizat & Bach, 2018). Thus GN with adaptive damping mitigates the kernel-conditioning slowdown in the convergence rate that hinders first-order optimization.

- $\alpha > 0$ (output scaling) and the network width $m$ are design parameters; our guarantees in the overparameterized regime hold provided $\alpha\sqrt{m}$ is sufficiently large.

Computation of $\lambda_{\min}(\boldsymbol{K}_t)$ for the damping factor $\rho(w_t)$ can be impractical. As such, a provably good time-constant damping factor is desirable. The following corollary to Theorem 3.4 identifies such a practical constant and data-dependent damping factor with a similar bound.

**Corollary 3.6** (Convergence of GN – constant damping). *The Gauss–Newton gradient flow (4) with a constant damping factor* $\rho_t = \rho \in (0, \frac{\lambda^2}{1+\lambda^2}]$, $t \in [0,\infty)$ *yields*

$$V_t \leq V_0 \cdot \exp\left(-\frac{2\nu t\lambda^2}{\rho + (1-\rho)\lambda^2}\right),$$

*under Assumption 3.1 when*

$$\alpha\sqrt{m} \geq \frac{\sigma_2\mu X\sqrt{nV_0}}{\nu^{3/2}}\frac{1}{\lambda^2}.$$

*In particular,* $\rho_t = \rho^{\mathrm{con}}$ *where*

$$\rho^{\mathrm{con}} := \frac{\lambda^2}{1+\lambda^2}, \tag{11}$$

*yields*

$$V_t \leq V_0 e^{-\nu(1+\lambda^2)t} \leq V_0\, e^{-\nu t}$$

*for all* $t \geq 0$.

The proof of Corollary 3.6 is given in Appendix A.

We next show that the discrete-time Gauss–Newton method with appropriately chosen step size and damping factor achieves a geometric convergence rate *independent* of $\lambda^2$.

**Theorem 3.7** (Convergence of GN – discrete time). *Let*

$$h_i(z) := \frac{z^2}{\left((1-\rho)z^2 + \rho\right)^i}, \quad i = 1, 2,$$

*and* $\alpha = 1$. *Consider the following discrete-time Gauss–Newton method:*

$$w_{k+1} = w_k - \eta\left[\boldsymbol{H}_\rho^{\mathrm{G}}(w_k)\right]^{-1}\mathrm{D}^\top f(w_k)\nabla g(f(w_k)).$$

*Under Assumption 3.1, the damping factor* $\rho \in \left(0, \frac{\lambda^2}{1+\lambda^2}\right]$ *and the constant step size* $\eta \leq \frac{h_1(\lambda)}{6h_1^2(\mathrm{Lip}_f)\mu}$ *yields*

$$V_k \leq V_0\left(1 - \eta\nu h_1(\lambda)\right)^k \text{ for any } k \in \mathbb{N}, \tag{12}$$

*for any network width $m \in \mathbb{N}$ that satisfies*

$$\frac{\sigma_2 X \sqrt{n}}{\sqrt{m}} \leq \sqrt{\frac{\nu}{V_0}} \min \left\{ \frac{h_1(\mathrm{Lip}_f)}{h_2(\lambda)\mu\eta}, \frac{h_1^2(\mathrm{Lip}_f)}{h_2(\lambda)}, \frac{\lambda\nu h_1(\lambda)}{2\sqrt{2h_2(\lambda)}} \right\}.$$

*Setting $\eta = \frac{h_1(\lambda)}{6h_1^2(\mathrm{Lip}_f)\mu}$ and $\rho = \rho^{\mathrm{con}}$ yields*

$$V_k \leq V_0 \left( 1 - \frac{1}{24} \cdot \frac{\nu}{\mu} \cdot \frac{1}{h_1^2(\mathrm{Lip}_f)} \right)^k.$$

The proof of Theorem 3.7 can be found in Appendix A. For quadratic loss in the kernel regime, we characterize the implicit bias of Gauss–Newton dynamics of the linearized model in Appendix C.

### 3.1.2. GENERALIZED GAUSS–NEWTON: LEVERAGING CURVATURE

The Gauss–Newton preconditioner $\boldsymbol{H}_\rho^{\mathrm{G}}$ avoids computing $\nabla^2 g$ and yields global geometric convergence with bounds that are robust to ill-conditioned kernel matrices. In the following, we analyze the generalized Gauss–Newton preconditioner $\boldsymbol{H}_\rho$, which incorporates the loss curvature through $\nabla^2 g$, and establish a global convergence guarantee under a curvature-aware damping schedule.

We define the following curvature-dependent damping schedule:

$$\rho^{\mathrm{glo}}(w) := \frac{\lambda^2 \cdot \lambda_{\max}\big(\nabla^2 g(\alpha f(w))\big)}{1 + \lambda^2 \cdot \lambda_{\max}\big(\nabla^2 g(\alpha f(w))\big)}. \quad (13)$$

We next establish a global convergence guarantee for GGN under the curvature-dependent damping $\rho^{\mathrm{glo}}$.

**Theorem 3.8** (Convergence of generalized GN). *Consider the preconditioned gradient flow* (4) *with $\boldsymbol{H} = \boldsymbol{H}_\rho$ and $\alpha = 1$. Under Assumption 3.1, suppose*

$$\sqrt{m} \geq \frac{3\sigma_2}{\sqrt{\nu}} \frac{\lambda^2 + \nu^{-1}}{\lambda^2 + \mu^{-1}} \frac{\mu}{\lambda^2} X \sqrt{n}.$$

*If $\rho_t = \rho^{\mathrm{glo}}(w_t)$, then for all $t \geq 0$,*

$$V_t \leq V_0 \exp\left( -\nu\lambda^2 t - \nu \int_0^t \lambda_{\max}^{-1}\big(\nabla^2 g(f(w_s))\big) \mathrm{d}s \right)$$

*Moreover, there exists $w^\star \in \mathcal{B}(w_0, r_0)$ such that*

$$\lim_{t \to \infty} w_t = w^\star \quad \text{and} \quad f(w^\star) \in \arg\min_{\xi \in \mathbb{R}^n} g(\xi). \quad (14)$$

*Remark* 3.9. Theorem 3.8 can be read as a globalization principle for Gauss–Newton. Classical Newton-type methods achieve fast convergence only locally, once the iterates enter a neighborhood where the second-order model is accurate and the Hessian is well-conditioned (Nocedal & Wright,

1999). Here, the loss-dependent damping $\rho^{\mathrm{glo}}$ acts as an algorithmic regularizer that keeps the preconditioner globally well-behaved and guarantees global progress toward an interpolating limit. If, additionally, $w \mapsto \nabla^2\mathcal{R}(w)$ is $C$-Lipschitz and $\nabla^2\mathcal{R}(w^\star) \succ 0$, which may hold if $p = n$, after the trajectory enters a neighborhood of $w^\star$, switching to a particular damping schedule makes $\boldsymbol{H}_\rho(w)$ a controlled perturbation of the Hessian $\nabla^2\mathcal{R}$, thereby yielding convergence rates independent of also $\nu$ by local Newton-type arguments (Dennis & Moré, 1974). A discussion of this phenomenon can be found in Appendix A.

### 3.2. Gauss–Newton for Deep Networks

We extend the overparameterized analysis to deep networks where the output layer $c$ is trained. The main technical challenge is that the map $w \mapsto f(w)$ is not globally Lipschitz-smooth. We handle this by adapting the stability tools proposed in (Du et al., 2019) with slightly sharper dependence on the sample size $n$ and confidence $\delta$.

In this subsection (only), we also assume that $x_i \in \mathbb{S}^{d-1}$, $\sup_{z \in \mathbb{R}} |\sigma(z)| \leq \sigma_0 < \infty$, which holds for tanh and sigmoid. We also assume that the top-layer kernel satisfies $\boldsymbol{K}^{(H)}(w_0) \succeq 4\lambda^2 I$ for some $\lambda > 0$. By Lemma B.2 of (Du et al., 2019), this condition holds for sufficiently large width $m$ whenever $x_i \nparallel x_j$ for all $i \neq j$. Define

$$\boldsymbol{K}^{(h)}(w) := \mathrm{D}_h f(w) \, \mathrm{D}_h^\top f(w)$$

and $\boldsymbol{K}(w) := \sum_{h=1}^H \boldsymbol{K}^{(h)}(w)$, so $\boldsymbol{K}(w) \succeq \boldsymbol{K}^{(H)}(w)$ for all $w$.

**Lemma 3.10.** *Let $C = 2\sigma_0\sigma_1 a_\sigma$. For any $\delta \in (0, 1)$, let*

$$m \geq \sigma_0^4 a_\sigma^2 \left( \frac{C^H - 1}{2(C-1)} \right)^2 \log\left( \frac{2Hn}{\delta} \right), \quad (15)$$

*and $k_w := 1 + 2\sqrt{\max\{1, d/m\}}$. Then, with probability at least $1 - \delta - 2He^{-m/2}$ over the random initialization, if*

$$\|w - w_0\|_2 \leq R\sqrt{m},$$

*where $R := k\frac{k_x - 1}{k_x^H - 1} \min\{1, n^{-1}\lambda_{\min}(\boldsymbol{K}_0^{(H)})\}$ for a universal constant $k > 0$, we have*

$$\lambda_{\min}(\boldsymbol{K}(w)) \geq \lambda_{\min}(\boldsymbol{K}^{(H)}(w)) \geq \lambda_{\min}(\boldsymbol{K}^{(H)}(w_0))/4.$$

Lemma 3.10 refines Lemma B.4 in (Du et al., 2019) with slightly improved dependencies on $n$ and $\delta$, and its proof can be found in Appendix A. Using Lemma 3.10, we obtain the following convergence result.

**Corollary 3.11** (Convergence of GN – deep networks). *For any $\delta \in (0, 1)$, if the neural network width $m$ is chosen sufficiently large such that* (15) *holds, then the Gauss–Newton flow with a constant damping $\rho \in (0, \frac{\lambda^2}{1+\lambda^2}]$ and*

$\alpha\sqrt{m} \geq \frac{\mu\sqrt{2V_0}}{\lambda R \nu^{3/2}}$ *yields*

$$V_t \leq V_0 \exp\left(-\frac{2\nu\lambda^2 t}{\rho + (1-\rho)\lambda^2}\right), \quad t \geq 0$$

*with probability at least $1 - \delta - 2He^{-m/2}$ over the random initialization.*

The constant step-size choice $\rho^{\text{con}}$ in (11) yields a $\lambda$-independent upper bound for the optimality gap similar to Corollary 3.6.

## 4. Riemannian Geometry and Convergence in the Underparameterized Regime

In this section, we consider the underparameterized regime $(p < n)$ for two-layer $(H = 1)$ networks with smooth activations in a similar setting as Section 3.1. The kernel matrix $\boldsymbol{K}_t \in \mathbb{R}^{n \times n}$ is singular since $\text{rank}(Df(w_t)D^\top f(w_t)) \leq p < n$ for all $t \in \mathbb{R}_+$. As a result, the analyses in the preceding section, which relies on the non-singularity of $\boldsymbol{K}_t$, do not extend to this setting. To address this, we utilize tools from optimization on Riemannian manifolds. In this regime, it is natural to work with the (undamped) Gauss–Newton preconditioner $\boldsymbol{H}(w) = Df(w)^\top Df(w)$, thus we consider the Gauss–Newton flow (4). The non-singularity of $\boldsymbol{H}(w_0)$ under random initialization $w_0$ will be crucial in establishing the Riemannian optimization framework in the sequel.

**Assumption 4.1.** There exists $\lambda_0 > 0$ with $\boldsymbol{H}(w_0) \succcurlyeq 4\lambda_0^2 \boldsymbol{I}$.
Let

$$B := \left\{ w \in \mathbb{R}^p : \|w - w_0\|_2 \leq r_0 \right\},$$

where

$$r_0 = \min\left\{ \frac{\lambda_0}{L}, \frac{1}{4} \cdot \frac{\lambda_0^2 \nu}{\mu L \text{Lip}_f} \right\}. \tag{16}$$

The following result implies the non-degeneracy of the Gram matrix $\boldsymbol{H}(w)$ on $B$.

**Lemma 4.2** ($w \mapsto \alpha f(w)$ is an immersion on $B$)**.** *For any $w \in B$, we have*

$$\boldsymbol{H}(w) = D^\top f(w) Df(w) \succcurlyeq \lambda_0^2 \boldsymbol{I},$$

*which implies that $\text{rank}(Df(w)) = p$ for all $w \in B$. Thus, $\alpha f$ is an immersion on $B$.*

The result follows from (Chizat et al., 2019), and we provide the proof in Appendix B for completeness.

Let

$$T := \inf\{t > 0 : w_t \notin B\}$$

be the first-exit time of $B$. Then, the Gauss–Newton gradient flow is well-defined for $t < T$ since we have a non-degenerate preconditioner with $\inf_{t<T} \lambda_{\min}(\boldsymbol{H}(w_t)) \geq \lambda_0^2$ by Lemma 4.2.

We first characterize the evolution in function space and the energy dissipation inequality in the underparameterized regime.

**Lemma 4.3.** *For any $w \in B$, let*

$$\boldsymbol{P}(\alpha f(w)) := Df(w)\big[\boldsymbol{H}(w)\big]^{-1}D^\top f(w). \tag{17}$$

*Then, for any $t < T$,*

$$\frac{d\alpha f(w_t)}{dt} = -\boldsymbol{P}(\alpha f(w_t))\nabla g(\alpha f(w_t)), \tag{GF-Ou}$$

$$\frac{dg(\alpha f(w_t))}{dt} = -\|\boldsymbol{P}(\alpha f(w_t))\nabla g(\alpha f(w_t))\|_2^2. \tag{EDI-u}$$

*Remark 4.4* (Why Riemannian optimization?)**.** For $t < T$, $\boldsymbol{P}(\alpha f(w_t))$ is symmetric and idempotent, hence an orthogonal projection, and $\text{rank}(\boldsymbol{P}(\alpha f(w_t))) = p < n$. As a result, (EDI-u) only implies that $t \mapsto g(\alpha f(w_t))$ is non-increasing, which does not yield a convergence rate or characterize the limiting predictor. This motivates us to cast the problem as an optimization problem on a Riemannian manifold.

An immediate question in studying (GF-Ou) is the characterization of the subspace that $\boldsymbol{P}(\alpha f(w_t))$ projects the Euclidean gradient $\nabla g(\alpha f(w_t))$ onto. This leads us to depart from Euclidean geometry and study $\alpha f(B)$ as a smooth submanifold.

For any $\alpha > 0$, let

$$\mathcal{M} := \alpha f(B) := \{\alpha f(w) : w \in B\}. \tag{18}$$

**Lemma 4.5.** $\alpha f|_B : B \to \mathcal{M}$ *is injective, and it is a smooth embedding on $B$.*

The following result shows that the function space $\mathcal{M}$ is a smooth embedded submanifold of the Euclidean space $\mathbb{R}^n$.

**Proposition 4.6.** $\mathcal{M}$ *is a $p$-dimensional smooth embedded submanifold of $\mathbb{R}^n$ with boundary.*

In the following, we show that $(\mathcal{M}, \langle \cdot, \cdot \rangle^{\mathcal{M}})$ is a Riemannian submanifold of the function space $\mathbb{R}^n$ of predictors.

**Lemma 4.7.** *For any $w \in B$, let*

$$\mathcal{T}_{\alpha f(w)}\mathcal{M} := \{\alpha Df(w)z : z \in \mathbb{R}^p\} = \text{Im}(Df(w)). \tag{19}$$

*Then, $\mathcal{T}_{\alpha f(w)}\mathcal{M}$ is the tangent space of $\alpha f(w) \in \mathcal{M}$. For any $w \in B$ and $u, v \in \mathcal{T}_{\alpha f(w)}\mathcal{M}$,*

$$\langle u, v \rangle^{\mathcal{M}}_{\alpha f(w)} := \langle u, v \rangle = u^\top v$$

*is a Riemannian metric on $\mathcal{M}$. Consequently, $\left(\mathcal{M}, \langle \cdot, \cdot \rangle^{\mathcal{M}}\right)$ is a Riemannian submanifold of $\mathbb{R}^n$.*

The following result shows that the Gauss–Newton dynamics in the underparameterized regime corresponds to a Riemannian gradient flow in the function space.

**Proposition 4.8** (GN as a Riemannian gradient flow). *For any $\alpha f(w) \in \mathcal{M}$, $\boldsymbol{P}(\alpha f(w))$ is the projection operator onto its tangent space $\mathcal{T}_{\alpha f(w)}\mathcal{M}$, i.e.,*

$$\boldsymbol{P}(\alpha f(w))z = \underset{y \in \mathcal{T}_{\alpha f(w)}\mathcal{M}}{\arg\min} \|y - z\|^2.$$

*Furthermore,*

$$\operatorname{grad}_{\alpha f(w)}^{\mathcal{M}} g(\alpha f(w)) := \boldsymbol{P}(\alpha f(w)) \nabla g(\alpha f(w)) \quad (20)$$

*is the Riemannian gradient of $g$ at $\alpha f(w) \in \mathcal{M}$. Consequently, the GN gradient flow induces*

$$\begin{aligned}
\frac{\mathrm{d}\alpha f(w_t)}{\mathrm{d}t} &= -\boldsymbol{P}(\alpha f(w_t)) \nabla g(\alpha f(w_t)) \\
&= -\operatorname{grad}_{\alpha f(w_t)}^{\mathcal{M}} g(\alpha f(w_t)),
\end{aligned}$$

*corresponds to Riemannian gradient flow on $(\mathcal{M}, \langle \cdot, \cdot \rangle^{\mathcal{M}})$.*

In Figure 1, we illustrate trajectories for a single-neuron (i.e., $m = 1$) with $\tanh$ activation function in the function and parameter spaces on a problem with $n = 3$ random data points of dimension $d = 2$. The embedded submanifold $\mathcal{M} = \alpha f(B)$ is the two-dimensional surface in the function space $\mathbb{R}^3$, as illustrated in Figure 1.

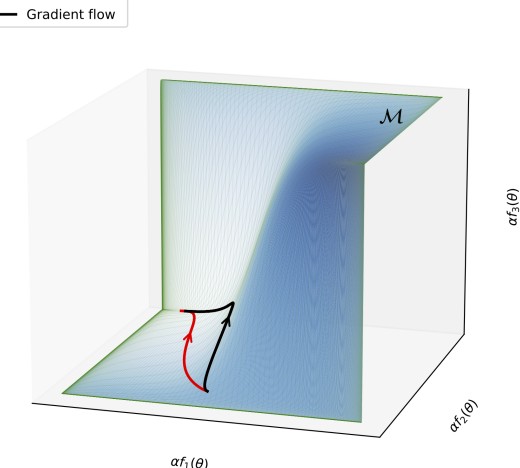

*Figure 1.* Trajectories of Gauss–Newton and gradient flow in function space for $n = 3$ and $p = 2$ in nonlinear regression. Gauss–Newton induces Riemannian gradient flow on $\mathcal{M}$.

**Lemma 4.9.** *Any two points of $\mathcal{M}$ are joined by a length-minimizing curve in $\mathcal{M}$ with respect to the intrinsic Riemannian distance.*

Lemma 4.9 implies that for any points $\alpha f(w), \alpha f(w') \in \mathcal{M}$, there exists a length-minimizing curve $c : [0, 1] \to \mathcal{M}$ such that $c(0) = \alpha f(w)$ and $c(1) = \alpha f(w')$. The complete proof can be found in Appendix B.

In the following, we establish key geodesic regularity properties for the loss function on the manifold $\mathcal{M}$, which constitute an essential part of the convergence analysis. To that end, by the same argument as in Lemma 4.5, $\mathcal{M}_\epsilon := \alpha f(B_\epsilon)$ for $B_\epsilon := \{w \in \mathbb{R}^p : \|w - w_0\|_2 < r_0 + \epsilon\}$ is a smooth embedded manifold *without* boundary containing $\mathcal{M}$ for any $\epsilon \in (0, \lambda_0/L)$. In the Riemannian arguments below, geodesics, exponential maps, Riemannian Hessians and parallel transports are taken on the boundaryless manifold $\mathcal{M}_\epsilon \supset \mathcal{M}$. In order to use tools from Riemannian optimization (Boumal, 2023), we impose the standing geometric condition that $\mathcal{M}$ is geodesically convex in $\mathcal{M}_\epsilon$ for an arbitrarily small $\epsilon \in (0, \lambda_0/L)$.

Let

$$\mathcal{S} := \{y \in \mathcal{M} : g(y) \le g(0)\},$$

which is a nonempty set since $g(\alpha f(w_0)) = g(0)$, thus $\alpha f(w_0) \in \mathcal{S}$. As a consequence of the energy dissipation inequality (EDI-u), $t \mapsto g(\alpha f(w_t))$ is a non-increasing function on $t < T$, thus under Gauss–Newton dynamics, the optimization trajectory lies in $\mathcal{S}$:

$$\{\alpha f(w_t) : t \in [0, T)\} \subset \mathcal{S} \subset \mathcal{M}.$$

**Theorem 4.10** (Geodesic regularity of $g$). *For any*

$$\alpha \ge \frac{4L\mu}{\nu\lambda_0^2} \max\left\{\|f^\star\|, \sqrt{\frac{2g(0)}{\nu}}\right\} =: \alpha_0, \quad (21)$$

*the following results hold:*

(a) *The loss function $g|_{\mathcal{M}} : \mathcal{M} \to \mathbb{R}$ is geodesically convex, i.e., for any $z \in \mathcal{M}, v \in \mathcal{T}_z\mathcal{M}$ and $c(t) = \operatorname{Exp}_z(vt)$, $t \in [0, 1]$, we have*

$$g(z) + t\langle \operatorname{grad}_z^{\mathcal{M}} g(z), v \rangle_z^{\mathcal{M}} \le g(\operatorname{Exp}_z(tv)),$$

*for any $t \in [0, 1]$.*

(b) *The sublevel set*

$$\mathcal{S} := \{z \in \mathcal{M} : g(z) \le g(0)\}$$

*is geodesically convex.*

(c) *$g|_{\mathcal{S}}$ is a $\frac{\nu}{2}$-geodesically strongly convex function on $\mathcal{S}$: for any $z \in \mathcal{S}, v \in \mathcal{T}_z\mathcal{M}$ and $c(t) = \operatorname{Exp}_z(vt)$ for $t \in [0, 1]$, we have*

$$g(z) + t\langle \operatorname{grad}_z^{\mathcal{M}} g(z), v \rangle_z^{\mathcal{M}} \le g(\operatorname{Exp}_z(tv)) - t^2 \frac{\nu}{4}\|v\|^2,$$

*for any $t \in [0, 1]$.*

(d) *$g|_{\mathcal{S}}$ has geodesically $\frac{3\mu}{2}$-Lipschitz continuous gradients: for any $z \in \mathcal{S}, v \in \mathcal{T}_z\mathcal{M}$ and $c(t) = \operatorname{Exp}_z(vt)$ for $t \in [0, 1]$, we have*

$$g(z) + t\langle \operatorname{grad}_z^{\mathcal{M}} g(z), v \rangle_z^{\mathcal{M}} \ge g(\operatorname{Exp}_z(tv)) - t^2 \frac{3\mu}{4}\|v\|_z^2,$$

*for any $t \in [0, 1]$.*

The proof of Theorem 4.10, which is provided in Appendix B, relies on establishing lower bounds on the minimum singular value of the Riemannian Hessian of $g$ on $\mathcal{M}$ and $\mathcal{S}$.

*Remark* 4.11 (Controlling the curvature by $\alpha$). A key step in the proof of Theorem 4.10 is to bound $\left\|\frac{d}{dt}\left[\boldsymbol{P}(\alpha f(\gamma_t))\right]\big|_{t=0}\right\|$ where $\gamma_t : [0,1] \to B$ is any smooth curve such that $\gamma_0 = w \in B$ and

$$\frac{d\alpha f(\gamma_t)}{dt}\bigg|_{t=0} = u \in \mathcal{T}_{\alpha f(w)}\mathcal{M}\backslash\{0\},$$

which corresponds to the magnitude of the second fundamental form along the direction $u$, and thus quantifies curvature. In order to establish the geodesic regularity of $g$, we choose $\alpha$ sufficiently large to control the curvature of $\mathcal{M}$.

Since $\mathcal{M}$ is a compact and smooth embedded Riemannian manifold, there exists an in-class optimal predictor $f^\star_\mathcal{M}$ such that

$$g(f^\star_\mathcal{M}) = \inf\{g(z) : z \in \mathcal{M}\}.$$

We make the following representational assumption regarding the existence of a critical point in the interior $\mathcal{M}$.

**Assumption 4.12.** There exists $w^\star_\alpha \in \text{int}(B)$ such that

$$\text{grad}^\mathcal{M}_{\alpha f(w^\star_\alpha)} g(\alpha f(w^\star_\alpha)) = 0.$$

The next lemma establishes a Riemannian Polyak–Lojasiewicz inequality for $g$ on $\mathcal{M}$, which we will use to prove exponential decay of the optimality gap $V_t$.

**Lemma 4.13.** *For any $\alpha \geq \alpha_0$ where $\alpha_0$ is given in (21), for all $t \in [0, T)$, there exists a tangent vector $v_t \in \mathcal{T}_{\alpha f(w^\star_\alpha)}\mathcal{M}$ such that*

$$\frac{\nu}{4}\|v_t\|_2^2 \leq V_t \leq \frac{1}{\nu}\|\text{grad}^\mathcal{M}_{\alpha f(w_t)}g(\alpha f(w_t))\|_2^2 \qquad (22)$$

$$\|\text{grad}^\mathcal{M}_{\alpha f(w_t)}g(\alpha f(w_t))\|_2 \leq \frac{3\mu}{2}\|v_t\|_2, \qquad (23)$$

*where*

$$V_t := g(\alpha f(w_t)) - g(\alpha f(w^\star_\alpha))$$

*for $t \geq 0$.*

Proof of Lemma 4.13 is given in Appendix B.

In the following, we present the main convergence and optimality result for the Gauss–Newton gradient flow in the underparameterized regime.

**Theorem 4.14** (Convergence of GN – underparameterized). *For the scaling parameter*

$$\alpha = \max\left\{\alpha_0, \frac{6\mu\sqrt{g(0)}}{\lambda_0 \nu^{\frac{3}{2}} r_0}\right\},$$

*the Gauss–Newton flow achieves*

$$V_t \leq V_0 \exp(-\nu t) \leq g(0)\exp(-\nu t)$$

*for any $t \in [0, \infty)$ under Assumptions 4.1-4.12. Furthermore,*

$$\|w_t - w_0\|_2 \leq r_0 \quad \text{and} \quad \boldsymbol{H}(w_t) \succeq \lambda_0^2 \boldsymbol{I} \qquad (24)$$

*for all $t \in \mathbb{R}^+$.*

The proof of Theorem 4.14 is given in Appendix B.

*Remark* 4.15 (Benefits of preconditioning in the underparameterized regime). We have the following observations on the superiority of the Gauss–Newton gradient flow in the underparameterized regime compared to the gradient flow.

- **Exponential convergence rate for the last-iterate.** The Gauss–Newton gradient flow achieves *exponential* convergence rate $\exp(-\Omega(t))$ for the last-iterate in the underparameterized regime. The convergence rate for gradient descent in this regime is subexponential (Ji & Telgarsky, 2020; Cayci & Eryilmaz, 2025) under compatible assumptions.

- **Convergence without explicit regularization.** The convergence result in Theorem 4.14 holds *without* any explicit regularization scheme, e.g., early stopping or projection. The Gauss–Newton gradient flow converges in the underparameterized setting as in (24). In the underparameterized regime, the gradient descent dynamics requires an explicit regularization scheme to control the parameter movement $\|w_t - w_0\|$, e.g., early stopping (Ji & Telgarsky, 2020; Cayci & Eryilmaz, 2025) or projection (Cai et al., 2019a; Cayci et al., 2023; Cayci & Eryilmaz, 2025).

- **$\lambda_0$-independent convergence rate.** The convergence rate in Theorem 4.14 is independent of the minimum eigenvalue $\lambda_0$ of the Gram matrix $\text{D}^\top f \text{D} f$, which indicates that the performance of the Gauss–Newton dynamics is resilient against ill-conditioned Gram matrices due to the geometry of the input data points $\{x_j\}_{j=1,\ldots,n} \subset \mathbb{R}^d$.

## 5. Conclusions

In this work, we analyzed the Gauss–Newton dynamics for underparameterized and overparameterized neural networks in the near-initialization regime, and demonstrated that the recent optimization tools developed for embedded submanifolds can provide important insights into the training dynamics of neural networks. As a follow-up to this work, the performance analysis of the Gauss–Newton method in the mean-field regime (Chizat & Bach, 2018; Mei et al., 2019; Sirignano & Spiliopoulos, 2020) is an interesting future direction.

## Impact Statement

This paper presents work whose goal is to advance the field of Machine Learning. There are many potential societal consequences of our work, none which we feel must be specifically highlighted here.

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

## A. Omitted Proofs from Section 3

*Proof of Lemma 3.3.* By the chain rule, we obtain

$$\frac{\mathrm{d}\alpha f(w_t)}{\mathrm{d}t} = \alpha \mathrm{D}f(w_t)\frac{\mathrm{d}w_t}{\mathrm{d}t}$$
$$= -\mathrm{D}f(w_t)[\boldsymbol{H}_\rho^{\mathrm{G}}(w_t)]^{-1}\mathrm{D}^\top f(w_t)\nabla g(\alpha f(w_t)).$$

Since $t < T$, the empirical kernel matrix $\boldsymbol{K}_t$ is non-singular. Thus, by applying the Sherman-Morrison-Woodbury matrix identity (Horn & Johnson, 2012) to the above, we obtain

$$\mathrm{D}f(w_t)[\boldsymbol{H}_\rho^{\mathrm{G}}(w_t)]^{-1}\mathrm{D}^\top f(w_t)$$
$$= \frac{1}{\rho_t}\mathrm{D}f(w_t)\left[\boldsymbol{I} - \frac{1-\rho_t}{\rho_t}\mathrm{D}f(w_t)\left[\boldsymbol{I} + \frac{1-\rho_t}{\rho_t}\mathrm{D}f(w_t)\mathrm{D}^\top f(w_t)\right]^{-1}\mathrm{D}f(w_t)\right]\mathrm{D}^\top f(w_t)$$
$$= \frac{1}{\rho_t}\left(\boldsymbol{K}_t - \frac{1-\rho_t}{\rho_t}\boldsymbol{K}_t\left(\boldsymbol{I} + \frac{1-\rho_t}{\rho_t}\boldsymbol{K}_t\right)^{-1}\boldsymbol{K}_t\right).$$

This gives the gradient flow in the output space (GF-O).

For the energy dissipation inequality (EDI), first note that we have

$$\frac{\mathrm{d}g(\alpha f(w_t))}{\mathrm{d}t} = \frac{\mathrm{d}V_t}{\mathrm{d}t}$$
$$= \nabla^\top g(\alpha f(w_t))\frac{\mathrm{d}\alpha f(w_t)}{\mathrm{d}t}$$
$$= -\|\nabla g(\alpha f(w_t))\|^2_{\mathrm{D}f(w_t)[\boldsymbol{H}_\rho^{\mathrm{G}}(w_t)]^{-1}\mathrm{D}^\top f(w_t)} \tag{25}$$

where the last identity comes from (GF-O). As such, we need to characterize the spectrum, particularly the minimum singular value of $\boldsymbol{K}_t - \frac{1-\rho_t}{\rho_t}\boldsymbol{K}_t\left(\boldsymbol{I} + \frac{1-\rho_t}{\rho_t}\boldsymbol{K}_t\right)^{-1}\boldsymbol{K}_t$. To that end, let $(\gamma, u) \in \mathbb{R} \times \mathbb{R}^n$ be any eigenvalue-eigenvector pair for the matrix $\boldsymbol{K}_t$. Then,

$$\left(\boldsymbol{K}_t - \frac{1-\rho_t}{\rho_t}\boldsymbol{K}_t\left(\boldsymbol{I} + \frac{1-\rho_t}{\rho_t}\boldsymbol{K}_t\right)^{-1}\boldsymbol{K}_t\right)u = \gamma u - \frac{1-\rho_t}{\rho_t}\frac{\gamma^2}{1 + \frac{1-\rho_t}{\rho_t}\gamma}u$$
$$= \frac{\gamma\frac{\rho_t}{1-\rho_t}}{\frac{\rho_t}{1-\rho_t} + \gamma}u,$$

which implies that $\left(\frac{\gamma\rho_t}{(1-\rho_t)\gamma+\rho_t}, u\right)$ is an eigenpair for $\boldsymbol{K}_t - \frac{1-\rho_t}{\rho_t}\boldsymbol{K}_t\left(\boldsymbol{I} + \frac{1-\rho_t}{\rho_t}\boldsymbol{K}_t\right)^{-1}\boldsymbol{K}_t$, and therefore $\mathrm{D}f(w_t)[\boldsymbol{H}_\rho^{\mathrm{G}}(w_t)]^{-1}\mathrm{D}^\top f(w_t)$ has a corresponding eigenpair $\left(\frac{\gamma}{(1-\rho_t)\gamma+\rho_t}, u\right)$. Then, for any $(\rho_t)_{t\geq0}$ with $\inf_{t\geq0}\rho_t > 0$:

$$\|\nabla g(\alpha f(w_t))\|^2_{\mathrm{D}f(w_t)[\boldsymbol{H}_\rho^{\mathrm{G}}(w_t)]^{-1}\mathrm{D}^\top f(w_t)} \geq \|\nabla g(\alpha f(w_t))\|^2_2 \cdot \frac{\lambda_t^2}{(1-\rho_t)\lambda_t^2 + \rho_t}. \tag{26}$$

Substituting (26) into (25) concludes the proof of Lemma 3.3. □

**Lemma A.1.** *Under a constant damping scheme $\rho_t = \rho \in (0, 1]$, we have*

$$V_t \leq V_0 \exp\left(\frac{-2\nu\lambda^2 t}{\rho + (1-\rho)\lambda^2}\right),$$
$$\|\alpha f(w_t) - f^\star\|_2^2 \leq \frac{2V_0}{\nu}\exp\left(\frac{-2\nu t\lambda^2}{\rho + (1-\rho)\lambda^2}\right), \tag{27}$$

*for any $t \in [0, T)$, where*

$$V_t := g(\alpha f(w_t)) - g(f^\star)$$

*is the optimality gap, and $f^\star$ is the unique global minimizer of $g$ in $\mathbb{R}^n$.*

*Proof of Lemma A.1.* Note that $\frac{\mathrm{d}V_t}{\mathrm{d}t} = \frac{\mathrm{d}g(\alpha f(w_t))}{\mathrm{d}t}$, thus (EDI) with $\rho_t > 0$ implies

$$\frac{\mathrm{d}V_t}{\mathrm{d}t} \leq -\frac{\lambda_t^2}{(1-\rho_t)\lambda_t^2 + \rho_t}\|\nabla g(\alpha f(w_t))\|_2^2. \tag{28}$$

Since $\xi \mapsto g(\xi)$ is $\nu$-strongly convex, by Polyak-Lojasiewicz (PL) inequality (Bach, 2024), we have

$$\|\nabla g(\alpha f(w_t))\|_2^2 \geq 2\nu V_t.$$

Substituting this outcome of the PL-inequality and (26) into (25), we obtain

$$\frac{\mathrm{d}V_t}{\mathrm{d}t} \leq -\frac{\lambda_t^2}{(1-\rho_t)\lambda_t^2 + \rho_t} \cdot 2\nu V_t, \quad t \in [0, T). \tag{29}$$

Note that

$$g(f^\star) \leq g(\alpha f(w_t)) - \frac{\nu}{2}\|\alpha f(w_t) - f^\star\|_2^2, \tag{30}$$

since $f^\star \in \underset{x \in \mathbb{R}^n}{\arg\min}\, g(x)$ is the unique minimizer of the strongly convex $g : \mathbb{R}^n \to \mathbb{R}^+$, which implies that $\nabla g(f^\star) = 0$ by the first-order condition for optimality (Boyd & Vandenberghe, 2004). Thus,

$$\|\alpha f(w_t) - f^\star\|_2^2 \leq \frac{2V_t}{\nu} \tag{31}$$

for any $t < T$.

1. **Adaptive damping factor.** Consider

$$\rho_t = \frac{\lambda_t^2}{1 + \lambda_t^2}, \quad t < T.$$

   Since $\lambda_t^2 \geq \lambda^2 > 0$ for any $t < T$ by (8), we have $\rho_t > 0$. Substituting $\rho_t$ into (29), we obtain

$$\dot{V}_t \leq -\nu\left(1 + \lambda_t^2\right)V_t \leq -\nu\left(1 + \lambda^2\right)V_t \tag{32}$$

   since $\lambda_t \geq \lambda$ for all $t < T$. By Grönwall's lemma (Terrell, 2009), we obtain

$$V_t \leq V_0 \exp(-\nu(1 + \lambda^2)t), \quad t < T.$$

   Using the above inequality in (31), we conclude that

$$\|\alpha f(w_t) - f^\star\|_2^2 \leq \frac{2V_0}{\nu}\exp(-\nu(1 + \lambda^2)t), \quad t < T.$$

2. **Constant damping factor.** Consider $\rho_t = \rho \in (0, 1]$. Since $\lambda_t^2 \geq \lambda^2$ and $z \mapsto \frac{z}{(1-\rho)z+\rho}$ is a monotonically increasing function for any $\rho \geq 0$, (29) implies

$$\dot{V}_t \leq -\frac{\lambda^2}{(1-\rho)\lambda^2 + \rho} \cdot 2\nu V_t, \quad t < T.$$

   Thus, by Grönwall's lemma (Terrell, 2009), we obtain

$$V_t \leq V_0 \exp\left(-\frac{2\nu\lambda^2 t}{(1-\rho)\lambda^2 + \rho}\right) \tag{33}$$

   for any $t \in [0, T)$. Substituting (33) into (31) concludes the proof.

$$\square$$

*Proof of Theorem 3.4.* Note that the bound on the optimality gap $V_t$ in Lemma A.1 holds for $t < T$. Hence, in order to prove the convergence result, we will show that $T = \infty$. In the following, we will show that under the adaptive damping choice $\rho_t = \frac{\lambda_t^2}{1+\lambda_t^2}$ for $t \geq 0$, if

$$\alpha\sqrt{m} \geq \frac{\mu\sigma_2 X\sqrt{2V_0 n}}{\nu^{3/2}} \cdot \frac{1 + \lambda\mathrm{Lip}_f}{\lambda^2(1+\lambda^2)},$$

then $T = \infty$.

Note that we have $\lambda_s^2 \geq \lambda^2$ for any $s \in [0, T)$ by (8). Thus,

$$
\begin{aligned}
\|\nabla g(\alpha f(w_s))\|^2_{\mathrm{D}f(w_s)[\boldsymbol{H}_\rho^{\mathrm{G}}(\alpha f(w_s))]^{-2}\mathrm{D}^\top f(w_s)} &\leq \frac{\lambda_s^2}{((1-\rho_s)\lambda_s^2 + \rho_s)^2}\|\nabla g(\alpha f(w_s))\|_2^2 \\
&= \frac{1}{4} \cdot \frac{(1+\lambda_s^2)^2}{\lambda_s^2} \cdot \|\nabla g(\alpha f(w_s))\|_2^2 \\
&\leq \frac{1}{2} \cdot \frac{(1+\lambda_s^2)^2}{\lambda_s^2} \cdot \mu^2 \frac{V_0}{\nu} \exp\left(-\nu(1+\lambda^2)s\right).
\end{aligned}
$$

Hence, from (35), we have

$$
\begin{aligned}
\|w_t - w_0\|_2 &\leq \frac{1}{2\alpha}\left(\mathrm{Lip}_f + \frac{1}{\lambda}\right)\mu\sqrt{\frac{2V_0}{\nu}}\int_0^t \exp\left(-\nu s(1+\lambda^2)\right)\mathrm{d}s \\
&\leq \frac{1}{\alpha} \cdot \frac{1 + \lambda\mathrm{Lip}_f}{1+\lambda^2} \cdot \frac{1}{\lambda} \cdot \mu \cdot \sqrt{\frac{2V_0}{\nu^3}},
\end{aligned}
\tag{34}
$$

where we used $\lambda^2 \cdot \boldsymbol{I} \preccurlyeq \boldsymbol{K}_t \preccurlyeq \mathrm{Lip}_f^2 \cdot \boldsymbol{I}$. Note that $V_0 \leq g(0)$, where $g(0)$ is independent of $\alpha$, thus the upper bound in (34) can be controlled by choosing $\alpha > 0$ large. The choice of $\alpha$ given in the statement ensures that $\|w_t - w_0\| \leq r_0$. Thus, we conclude that $T = \infty$. $\qquad\square$

*Proof of Corollary 3.6.* The bounds on the optimality gap $V_t$ in Lemma A.1 holds for $t < T$. To prove convergence, we need to show that $T = \infty$. Consider the Gauss–Newton dynamics with any constant damping scheme $\rho_t = \rho \in \left(0, \frac{\lambda^2}{1+\lambda^2}\right]$ for $t \geq 0$. If $\alpha \geq \frac{\mu L \sqrt{2V_0}}{\nu^{3/2}}\frac{1}{\lambda^2}$, then $T = \infty$, as we will prove below.

By the triangle inequality, we have

$$
\begin{aligned}
\|w_t - w_0\|_2 = \left\|\int_0^t \dot{w}_s ds\right\|_2 &\leq \int_0^t \|\dot{w}_s\|_2 \mathrm{d}s \\
&= \frac{1}{\alpha}\int_0^t \|\nabla g(\alpha f(w_s))\|_{\mathrm{D}f(w_s)[\boldsymbol{H}_\rho^{\mathrm{G}}(\alpha f(w_s))]^{-2}\mathrm{D}^\top f(w_s)}\mathrm{d}s.
\end{aligned}
\tag{35}
$$

Using the Sherman-Morrison-Woodbury matrix identity (Horn & Johnson, 2012), we obtain

$$
\begin{aligned}
(1-\rho)^2 \mathrm{D}f(w_t)[\boldsymbol{H}_\rho^{\mathrm{G}}(w_t)]^{-2}\mathrm{D}^\top f(w_t) \\
= \rho_0^{-2}\boldsymbol{K}_t - 2\rho_0^{-3}\boldsymbol{K}_t(\boldsymbol{I} + \rho_0^{-1}\boldsymbol{K}_t)^{-1}\boldsymbol{K}_t + \rho_0^{-4}\boldsymbol{K}_t(\boldsymbol{I} + \rho_0^{-1}\boldsymbol{K}_t)^{-1}\boldsymbol{K}_t(\boldsymbol{I} + \rho_0^{-1}\boldsymbol{K}_t)^{-1}\boldsymbol{K}_t,
\end{aligned}
\tag{36}
$$

where $\rho_0 := \frac{\rho}{1-\rho}$ (Arbel et al., 2024). For any $z_1 \geq z_0 \geq \rho_0$, we have

$$\frac{z_1}{((1-\rho)z_1 + \rho)^2} \leq \frac{z_0}{((1-\rho)z_0 + \rho)^2}.\tag{37}$$

Then, if $(\gamma, u) \in \mathbb{R} \times \mathbb{R}^n$ is an eigenpair for $\boldsymbol{K}_t$, then $\left(\frac{\gamma}{((1-\rho)\gamma+\rho)^2}, u\right)$ is an eigenpair for the positive-definite matrix $\mathrm{D}f(w_t)[\boldsymbol{H}_\rho^{\mathrm{G}}(w_t)]^{-2}\mathrm{D}^\top f(w_t)$. Furthermore, by (37) and (8), the maximum eigenvalue of $\mathrm{D}f(w_t)[\boldsymbol{H}_\rho^{\mathrm{G}}(w_t)]^{-2}\mathrm{D}^\top f(w_t)$ is upper bounded by $\frac{\lambda^2}{((1-\rho)\lambda^2+\rho)^2}$. Thus, we have

$$\|\nabla g(\alpha f(w_s))\|^2_{\mathrm{D}f(w_s)[\boldsymbol{H}_\rho^{\mathrm{G}}(w_s)]^{-2}\mathrm{D}^\top f(w_s)} \leq \frac{\lambda^2}{((1-\rho)\lambda^2 + \rho)^2}\|\nabla g(\alpha f(w_s))\|_2^2,\tag{38}$$

for $s \le t < T$, which is implied by $\rho \le \frac{\lambda^2}{1+\lambda^2}$. Since $s \le t < T$, we have

$$
\begin{aligned}
\|\nabla g(\alpha f(w_s))\|_2 &= \|\nabla g(\alpha f(w_s)) - \nabla g(f^\star)\|_2 \\
&\le \mu \|\alpha f(w_s) - f^\star\| \\
&\le \mu \sqrt{\frac{2V_0}{\nu}} \exp\Big( -\frac{\nu\lambda^2 s}{\rho + (1-\rho)\lambda^2} \Big),
\end{aligned}
\tag{39}
$$

where the first line follows from the global optimality of $f^\star$, the second line is due to $\mu$-Lipschitz-continuous gradients of $g$, and the last line follows from the bound on the optimality gap in (31). Thus,

$$
\|\nabla g(\alpha f(w_s))\|_{\mathrm{D}f(w_s)[\boldsymbol{H}_\rho(\alpha f(w_s))]^{-2}\mathrm{D}^\top f(w_s)} \le \frac{\mu\sqrt{\frac{2V_0}{\nu}}\lambda}{(1-\rho)\lambda^2 + \rho} \exp\Big( -\frac{\nu\lambda^2 s}{\rho + (1-\rho)\lambda^2} \Big) \text{ for any } s < T.
$$

Substituting the above inequality into (35), we obtain

$$
\|w_t - w_0\|_2 \le \frac{1}{\alpha} \frac{\mu\sqrt{2V_0}}{\nu^{3/2}} \frac{1}{\lambda} \le r_0 := \frac{\lambda}{L}
$$

with the choice of $\alpha \ge \frac{\mu L\sqrt{2V_0}}{\nu^{3/2}} \frac{1}{\lambda^2}$. Recall that $L = \sigma_2 X\sqrt{\frac{n}{m}}$. Therefore, $\sup_{t<T} \|w_t - w_0\|_2 \le r_0$, where $r_0$ is independent of $T$, and hence $T = \infty$. $\qquad\square$

*Proof of Theorem 3.7.* The proof builds on the continuous time analysis of Theorem 3.4 and the discretization idea in (Du et al., 2018). For notational simplicity, let $\boldsymbol{D}_k := \mathrm{D}f(w_k)$, $\boldsymbol{H}_k := \boldsymbol{H}_\rho(w_k)$ for $k \in \mathbb{N}$.

Recall that $w \mapsto \mathrm{D}f(w)$ is $L$-Lipschitz with

$$
L = X\sigma_2\sqrt{\frac{n}{m}}.
$$

Let $N := \inf\{k \in \mathbb{N} : \|w_k - w_0\|_2 > \frac{\lambda_0}{L}\}$ and consider $k < N$. Since $g$ has $\mu$-Lipschitz gradients, we have

$$
g(f(w_{k+1})) \le g(f(w_k)) + \nabla^\top g(f(w_k))[f(w_{k+1}) - f(w_k)] + \frac{\mu}{2}\|f(w_{k+1}) - f(w_k)\|_2^2.
\tag{40}
$$

Since $w \mapsto f(w)$ has $L$-Lipschitz gradients, we have

$$
f(w_{k+1}) - f(w_k) = \boldsymbol{D}_k(w_{k+1} - w_k) + \boldsymbol{\epsilon}_k,
$$

where the local linearization error is bounded as

$$
\|\boldsymbol{\epsilon}_k\|_2 \le \frac{L}{2}\|w_{k+1} - w_k\|_2^2.
$$

Define

$$
h_i(z) = \frac{z^2}{\Big((1-\rho)z^2 + \rho\Big)^i} \text{ for } i = 1, 2.
\tag{41}
$$

Then, using (38), we have

$$
\begin{aligned}
\|w_{k+1} - w_k\|^2 &= \eta^2 \nabla^\top g(f(w_k)) \boldsymbol{D}_k \boldsymbol{H}_k^{-2} \boldsymbol{D}_k^\top \nabla g(f(w_k)) \\
&\le \eta^2 h_2(\lambda) \|\nabla g(f(w_k))\|^2 \\
&\le 2\eta^2 h_2(\lambda) \frac{\mu^2}{\nu} V_k,
\end{aligned}
\tag{42}
\tag{43}
$$

since $f \mapsto \nabla g(f)$ is $\mu$-Lipschitz and $\|f(w_k) - f^\star\|^2 \le \frac{2V_k}{\nu}$ by (31). Therefore,

$$
\|\boldsymbol{\epsilon}_k\| \le \frac{1}{2}\eta^2 L h_2(\lambda) \|\nabla g(f(w_k))\|^2 \le \eta^2 h_2(\lambda) \frac{\mu^2}{\nu} L V_k,
$$

and

$$\|f(w_{k+1}) - f(w_k)\|^2 \leq 2\|\boldsymbol{D}_k(w_{k+1} - w_k)\|^2 + 2\|\boldsymbol{\epsilon}_k\|^2$$

$$\leq 2\eta^2\|\boldsymbol{D}_k\boldsymbol{H}_k^{-1}\boldsymbol{D}_k^\top \nabla g(f(w_k))\|^2 + \eta^4 L^2 V_k h_2^2(\lambda)\frac{\mu^2}{\nu}\|\nabla g(f(w_k))\|^2. \tag{44}$$

Since $\|w_k - w_0\| \leq \frac{\lambda}{L}$, we have $\boldsymbol{D}_k\boldsymbol{D}_k^\top \preccurlyeq \mathrm{Lip}_f^2 \boldsymbol{I}$, which implies that $\boldsymbol{D}_k\boldsymbol{H}_k^{-1}\boldsymbol{D}_k^\top \preccurlyeq h_1(\mathrm{Lip}_f)$. Thus,

$$\|f(w_{k+1}) - f(w_k)\|^2 \leq \Big(2\eta^2 h_1^2(\mathrm{Lip}_f) + \eta^4 L^2 V_k h_2^2(\lambda)\frac{\mu^2}{\nu}\Big)\|\nabla g(f(w_k))\|^2. \tag{45}$$

On the other hand,

$$\nabla^\top g(f(w_k))\Big(f(w_{k+1}) - f(w_k)\Big) = \nabla^\top g(f(w_k))\Big(\boldsymbol{D}_k(w_{k+1} - w_k) + \boldsymbol{\epsilon}_k\Big).$$

Note that

$$\nabla^\top g(f(w_k))\boldsymbol{D}_k(w_{k+1} - w_k) = -\eta\nabla^\top g(f(w_k))\boldsymbol{D}_k\boldsymbol{H}_k^{-1}\boldsymbol{D}_k^\top \nabla g(f(w_k))$$

$$\leq -\eta h_1(\lambda)\|\nabla g(f(w_k))\|^2,$$

and

$$\nabla^\top g(f(w_k))\boldsymbol{\epsilon}_k \leq \|\nabla g(f(w_k))\| \cdot \|\boldsymbol{\epsilon}_k\|$$

$$\leq \eta^2\frac{\mu}{\sqrt{\nu}}L\sqrt{V_k}h_2(\lambda)\|\nabla g(f(w_k))\|_2^2.$$

Therefore, we have

$$\nabla^\top g(f(w_k))\Big(f(w_{k+1}) - f(w_k)\Big) \leq \Big(-\eta h_1(\lambda) + \eta^2\frac{\mu}{\sqrt{\nu}}L\sqrt{V_k}h_2(\lambda)\Big)\|\nabla g(f(w_k))\|^2. \tag{46}$$

Substituting (45) and (46) into (40), we obtain

$$V_{k+1} \leq V_k + \Big(-\eta h_1(\lambda) + \eta^2\frac{\mu}{\sqrt{\nu}}L\sqrt{V_k}h_2(\lambda) + \mu\eta^2 h_1^2(\mathrm{Lip}_f) + \eta^4 L^2 V_k h_2^2(\lambda)\frac{\mu^3}{\nu}\Big)\|\nabla g(f(w_k))\|^2.$$

Choose

$$\eta \leq \frac{h_1(\lambda)}{6\mu h_1^2(\mathrm{Lip}_f)},$$

and the network width $m$ sufficiently large such that $L$ satisfies

$$L \leq \sqrt{\frac{\nu}{V_0}}\min\left\{\frac{h_1(\mathrm{Lip}_f)}{h_2(\lambda)\mu\eta}, \frac{h_1^2(\mathrm{Lip}_f)}{h_2(\lambda)}\right\}. \tag{47}$$

Then, we have

$$V_{k+1} \leq V_k - \frac{\eta h_1(\lambda)}{2}\|\nabla g(f(w_k))\|^2$$

and $L\sqrt{V_k} \leq L\sqrt{V_0}$ for all $k \in \mathbb{N}$ by induction. From Polyak-Lojasiewicz inequality, we have $\|\nabla g(f(w_k))\|^2 \geq 2\nu V_k$. Using this, we obtain

$$V_{k+1} \leq \Big(1 - \eta\nu h_1(\lambda)\Big)V_k, \tag{48}$$

for any $k < N$. Hence, for any $k \leq N$,

$$V_k \leq V_0\Big(1 - \eta\nu h_1(\lambda)\Big)^k \quad \text{and} \quad \|f(w_k) - f^\star\|^2 \leq \frac{2V_0}{\nu}\Big(1 - \eta\nu h_1(\lambda)\Big)^k. \tag{49}$$

Using these inequalities, we will now show that $N = \infty$ can be established by sufficiently large $m \in \mathbb{N}$. First, recall that $\|w_{k+1} - w_k\|^2 \leq \eta^2 h_2(\lambda)\|\nabla g(f(w_k))\|^2$. Then, for $k < N$,

$$
\begin{aligned}
\|w_k - w_0\|_2 &\leq \sum_{s<k} \|w_{s+1} - w_s\|_2 \leq \eta\sqrt{h_2(\lambda)} \sum_{s<k} \|\nabla g(f(w_s))\|_2 \\
&\leq \eta\sqrt{h_2(\lambda)}\mu \sum_{s<k} \|f(w_s) - f^\star\|_2 \leq \eta\sqrt{h_2(\lambda)}\mu \sum_{s<k} \sqrt{\frac{2V_0}{\nu}} q^{s/2} \\
&\leq \eta\sqrt{h_2(\lambda)}\mu\sqrt{\frac{2V_0}{\nu}}\frac{1}{1-\sqrt{q}} \leq 2\eta\sqrt{h_2(\lambda)}\mu\sqrt{\frac{2V_0}{\nu}}\frac{1}{1-q} \\
&= \frac{2\sqrt{2h_2(\lambda)}}{\nu h_1(\lambda)}\sqrt{\frac{V_0}{\nu}},
\end{aligned}
$$

where $q := 1 - \eta\nu h_1(\lambda)$. We choose $m$ sufficiently large such that

$$
\frac{2\sqrt{2h_2(\lambda)}}{\nu h_1(\lambda)}\sqrt{\frac{V_0}{\nu}} \leq \frac{\lambda}{L}.
$$

Hence, we can ensure from the above inequality that $\|w_k - w_0\|_2 \leq r_0$ for all $k \in \mathbb{N}$, therefore $N = \infty$. Therefore, $V_k \leq V_0\left(1 - \eta\nu h_1(\lambda)\right)^k$ holds for any $k \in \mathbb{N}$, where we choose $m$ such that

$$
L \leq \sqrt{\frac{\nu}{V_0}}\min\left\{\frac{h_1(\mathrm{Lip}_f)}{h_2(\lambda)\mu\eta}, \frac{h_1^2(\mathrm{Lip}_f)}{h_2(\lambda)}, \frac{\lambda\nu h_1(\lambda)}{2\sqrt{2h_2(\lambda)}}\right\}. \tag{50}
$$

Choosing $\eta = \frac{h_1(\lambda)}{6\mu h_1^2(\mathrm{Lip}_f)}$ yields the convergence rate $V_k \leq V_0\left(1 - \frac{1}{6\kappa}\frac{h_1^2(\lambda)}{h_1^2(\mathrm{Lip}_f)}\right)^k$. $\qquad\square$

*Proof of Theorem 3.8.* Let $\boldsymbol{G}(w) := \nabla^2 g(\alpha f(w))$.

Using similar steps as Theorem 3.4 with Sherman-Morrison-Woodbury formula, the evolution in the function space can be written as

$$
\frac{\mathrm{d}f(w_t)}{\mathrm{d}t} = -\boldsymbol{\Sigma}_t(\boldsymbol{K}_t)\nabla g(f(w_t)),
$$

where $\boldsymbol{\Sigma}_t(\boldsymbol{K}) := \frac{1}{\rho_t}\left(\boldsymbol{K} - \bar{\rho}_t^{-1}\boldsymbol{K}\left(\boldsymbol{G}^{-1}(w_t) + \bar{\rho}_t^{-1}\boldsymbol{K}\right)^{-1}\boldsymbol{K}\right)$ with $\bar{\rho}_t = \rho_t/(1-\rho_t)$ for any symmetric positive semi-definite matrix $\boldsymbol{K} \in \mathbb{R}^{n\times n}$. The transform $\boldsymbol{K} \mapsto \boldsymbol{\Sigma}_t(\boldsymbol{K})$ is monotonically increasing with respect to the Löwner order. Since $t < T$, we have $\boldsymbol{K}_t \succeq \lambda^2 \boldsymbol{I}$, therefore $\boldsymbol{\Sigma}_t(\boldsymbol{K}_t) \succeq \boldsymbol{\Sigma}_t(\lambda^2\boldsymbol{I})$. Denote the eigenpairs of $\boldsymbol{G}(w_t)$ as $(u_{t,i}, \gamma_{t,i})$ where $\gamma_{t,i} \leq \gamma_{t,i+1}$ for all $i = 1, 2, \ldots, n-1$. Then, for any $i \in [n]$, $(u_i, \tilde{\gamma}_{t,i})$ is an eigenpair for $\boldsymbol{\Sigma}_t(\lambda^2\boldsymbol{I})$ where

$$
\tilde{\gamma}_{t,i} = \frac{\lambda^2}{\rho_t + (1-\rho_t)\lambda^2\gamma_{t,i}}.
$$

Using this, since $\gamma_{t,n} := \lambda_{\max}(\boldsymbol{G}(w_t))$, we conclude that

$$
\begin{aligned}
\frac{\mathrm{d}g(f(w_t))}{\mathrm{d}t} &\leq -\frac{\lambda^2}{\rho_t + (1-\rho_t)\lambda^2\gamma_{t,n}}\|\nabla g(f(w_t))\|^2 \\
&= -\frac{1}{2}\left(\lambda^2 + \frac{1}{\gamma_{t,n}}\right)\|\nabla g(f(w_t))\|^2, \\
&\leq -\nu(\lambda^2 + \gamma_{t,n}^{-1})(g(f(w_t)) - g(f^\star)) =: -\nu(\lambda^2 + \gamma_{t,n}^{-1})V_t,
\end{aligned}
$$

where the second line follows from the particular choice of $\rho_t$. Using Grönwall inequality, we obtain

$$
V_t \leq V_0\exp\left(-\lambda^2\nu t - \nu\int_0^t \frac{1}{\gamma_{s,n}}\mathrm{d}s\right),
$$

$$
\|f(w_t) - f^\star\| \leq \sqrt{\frac{2V_0}{\nu}}\exp(-\frac{1}{2}\lambda^2\nu t - \frac{1}{2}\frac{\nu}{\mu}t),
$$

for any $t < T$. Using $\|w_t - w_0\| \leq \int_0^t \|\dot{w}_s\| \mathrm{d}s$ similar to Theorem 3.4, we conclude that

$$\int_0^\infty \|\dot{w}_s\| \mathrm{d}s \leq Lc^\star,$$

for some constant $c^\star = c^\star(\mathcal{D}, \mu, \nu)$. Thus, if $m$ is sufficiently large as specified in the theorem statement, then

$$Lc^\star = \frac{\sigma_2 X \sqrt{n}}{\sqrt{m}} c^\star \leq r_0,$$

which implies that $T = \infty$. Note that $\int_0^\infty \|\dot{w}_s\| \mathrm{d}s < \infty$ implies that $\{w_t : t \geq 0\}$ is a Cauchy system in $\mathbb{R}^p$, thus $w_t \to w^\star$ as $t \to \infty$ for some $w^\star \in \mathcal{B}(w_0, r_0)$. Since $f(w_t) \underset{t \to \infty}{\to} f^\star$, the limit point should be an interpolant such that $f(w^\star) = f^\star$. $\qquad\square$

The following result, based on standard local Newton-type arguments (Dennis & Moré, 1974; Nocedal & Wright, 1999) and a damping schedule that does not require knowledge of $\|w - w^\star\|_2$, shows that the generalized Gauss–Newton achieves fast convergence rate locally around $w^\star$. However, this result requires $\nabla^2 \mathcal{R}(w^\star) \succ 0$, which can hold in the border case $p = n$. Without loss of generality, we normalize the minimum value to zero by replacing $g$ with $g(\cdot) - g(f^\star)$, so that $g(f^\star) = \min_\xi g(\xi) = 0$.

**Proposition A.2.** *Let $T_r := \inf\{t > 0 : \|w_t - w^\star\|_2 \leq r\}$ for $r > 0$. Assume $w \mapsto \nabla^2 \mathcal{R}(w)$ is $C$-Lipschitz continuous and $\nabla^2 \mathcal{R}(w^\star) \succ 0$. Then, there exists $r^\star := r^\star(C, \nabla^2 \mathcal{R}(w^\star))$ such that if we set $\rho_t = \rho^{\mathrm{loc}}(w_t)$ for $t \geq T_{r^\star}$, the generalized Gauss–Newton flow with $\rho_t = \rho^{\mathrm{loc}}(w_t)$ for the damping choice*

$$\rho^{\mathrm{loc}}(w) := \frac{\sqrt{\mathcal{R}(w)}}{\mathrm{Lip}_f \sqrt{\mu/2} + \sqrt{\mathcal{R}(w)}}$$

*satisfies*

$$\|w_t - w^\star\|_2^2 \leq \|w_{T_{r^\star}} - w^\star\|_2^2 \, e^{-\left(t - T_{r^\star}\right)}$$

*for all $t \geq T_{r^\star}$.*

*Proof.* We first make the following decomposition:

$$\boldsymbol{H}_\rho(w) - \nabla^2 \mathcal{R}(w^\star) = (1 - \rho(w)) \Big[ \underbrace{\mathrm{D}^\top f(w) \boldsymbol{G}(w) \mathrm{D}f(w) - \nabla^2 \mathcal{R}(w)}_{(a)} \Big]$$
$$+ (1 - \rho(w)) \Big[ \underbrace{\nabla^2 \mathcal{R}(w) - \nabla^2 \mathcal{R}(w^\star)}_{(b)} \Big] \qquad (51)$$
$$+ \rho(w) \Big[ \boldsymbol{I} - \nabla^2 \mathcal{R}(w^\star) \Big].$$

For $(a)$, we have the following identity:

$$\nabla^2 \mathcal{R}(w) - \mathrm{D}^\top f(w) \boldsymbol{G}(w) \mathrm{D}f(w) = \sum_{j=1}^n [\nabla g(f(w))]_j \nabla^2 \varphi(x_j; w).$$

Since $\|\nabla^2 \varphi(x_j; w)\| \leq \sigma_2 \|x_j\|^2 / \sqrt{m} \leq \sigma_2 X / \sqrt{m}$ and $L = X\sigma_2 \sqrt{n/m}$, Cauchy–Schwarz inequality yields

$$\|\nabla^2 \mathcal{R}(w) - \mathrm{D}^\top f(w) \boldsymbol{G}(w) \mathrm{D}f(w)\| \leq \mu \cdot L \cdot \mathrm{Lip}_f \cdot \|w - w^\star\|. \qquad (52)$$

For $(b)$, $C$-Lipschitz continuity of $\nabla^2 \mathcal{R}(w)$ implies

$$\|\nabla^2 \mathcal{R}(w) - \nabla^2 \mathcal{R}(w^\star)\| \leq C\|w - w^\star\|.$$

Also, $g(f(w)) \leq \frac{\mu}{2}\|f(w) - f(w^\star)\|^2 \leq \frac{\mu}{2}\mathrm{Lip}_f^2\|w - w^\star\|^2$, hence $\sqrt{\mathcal{R}(w)} \leq \mathrm{Lip}_f\sqrt{\mu/2}\|w_t - w^\star\|$, and

$$\frac{\sqrt{\mathcal{R}(w)}}{\mathrm{Lip}_f\sqrt{\mu/2} + \sqrt{\mathcal{R}(w)}} \leq \frac{\|w - w^\star\|}{1 + \|w - w^\star\|} \leq \|w - w^\star\|.$$

Using these, we obtain the following bound:

$$\|\boldsymbol{H}_\rho(w) - \nabla^2\mathcal{R}(w^\star)\| \leq \left(\mu \cdot L \cdot \mathrm{Lip}_f + C + 1 + \|\nabla^2\mathcal{R}(w^\star)\|\right)\|w - w^\star\| =: C'\|w - w^\star\|. \tag{53}$$

By Weyl's inequality (Horn & Johnson, 2012), we conclude that

$$\|w - w^\star\| \leq \frac{1}{2C'}\lambda_{\min}(\nabla^2\mathcal{R}(w^\star)) \implies \lambda_{\min}(\boldsymbol{H}_\rho(w)) \geq \frac{1}{2}\lambda_{\min}(\nabla^2\mathcal{R}(w^\star)). \tag{54}$$

For any $w \in \mathbb{R}^p$, let

$$e(w) := w - w^\star, \quad \text{and} \quad \psi(w) := \nabla\mathcal{R}(w) - \nabla^2\mathcal{R}(w^\star)e(w).$$

Using $C$-Lipschitz continuity of $\nabla^2\mathcal{R}(w)$,

$$\|\psi(w)\| \leq \frac{1}{2}C\|e(w)\|^2.$$

Then, letting $e_t := e(w_t)$, we obtain

$$\begin{aligned}
\dot{e}_t = \dot{w}_t &= -\boldsymbol{H}_\rho^{-1}(w_t)\nabla_w\mathcal{R}(w_t) \\
&= -\boldsymbol{H}_\rho^{-1}(w_t)\nabla^2\mathcal{R}(w^\star)e_t - \boldsymbol{H}_\rho^{-1}(w_t)\psi(w_t) \\
&= -e_t + \left([\nabla^2\mathcal{R}(w^\star)]^{-1} - \boldsymbol{H}_\rho^{-1}(w_t)\right)\nabla^2\mathcal{R}(w^\star)e_t - \boldsymbol{H}_\rho^{-1}(w_t)\psi(w_t).
\end{aligned}$$

We use the Lyapunov function $\Psi(w) := \frac{1}{2}\|e(w)\|_2^2$. Then, we have

$$\begin{aligned}
\frac{\mathrm{d}\Psi(w_t)}{\mathrm{d}t} &= e_t^\top \dot{e}_t \\
&\leq -\|e_t\|^2 + \|[\nabla^2\mathcal{R}(w^\star)]^{-1} - \boldsymbol{H}_\rho^{-1}(w_t)\| \cdot \lambda_{\max}(\nabla^2\mathcal{R}(w^\star))\|e_t\|^2 + \frac{C}{\lambda_{\min}(\nabla^2\mathcal{R}(w^\star))}\|e_t\|^3.
\end{aligned}$$

Using $\|A^{-1} - B^{-1}\| \leq \frac{\|A-B\|}{\lambda_{\min}(A)\lambda_{\min}(B)}$, if $\|w_t - w^\star\| \leq \frac{1}{2C'}\lambda_{\min}(\nabla^2(\mathcal{R}(w^\star)))$, we obtain

$$\|[\nabla^2\mathcal{R}(w^\star)]^{-1} - \boldsymbol{H}_\rho^{-1}(w_t)\| \leq \frac{2C'}{\lambda_{\min}^2(\nabla^2\mathcal{R}(w^\star))}\|e_t\|.$$

Hence, with $C'' := \frac{2C'\lambda_{\max}(\nabla^2\mathcal{R}(w^\star))}{\lambda_{\min}^2(\nabla^2\mathcal{R}(w^\star))} + \frac{C}{\lambda_{\min}(\nabla^2\mathcal{R}(w^\star))}$, we have

$$\frac{\mathrm{d}\Psi(w_t)}{\mathrm{d}t} \leq -2\Psi(w_t) + C''[\Psi(w_t)]^{3/2}.$$

If

$$\|w_t - w^\star\| \leq \min\left\{\frac{\sqrt{2}}{C''}, \frac{1}{2C'}\lambda_{\min}(\nabla^2\mathcal{R}(w^\star))\right\} =: r^\star, \tag{55}$$

then $C''[\Psi(w_t)]^{1/2} \leq 1$, and thus

$$\frac{\mathrm{d}\Psi(w_t)}{\mathrm{d}t} \leq -\Psi(w_t), \quad t \geq T(r^\star).$$

We conclude the proof by using Grönwall's lemma. $\qquad\square$

*Proof of Lemma 3.10.* The proof builds on and refines Lemma B.4 in (Du et al., 2019) with a slightly improved dependence on $n$ and $\delta$ for smooth and bounded activations. Let $\mathcal{F}_h = \sigma(W_0^{(1)}, \ldots, W_0^{(h)})$ for any $h \in [H]$. For notational simplicity, we denote $\boldsymbol{x}_j^{(h)}(\boldsymbol{W}_0)$ as $\boldsymbol{x}_j^{(h)}$ throughout the proof. For any data point $j \in [n]$,

$$\|\boldsymbol{x}_j^{(h)}\|_2^2 = \frac{a_\sigma}{m} \sum_{i=1}^{m} \sigma^2\big(\langle W_{0,i}^{(h)}, \boldsymbol{x}_j^{(h-1)}\rangle\big),$$

where $W_{0,i}^{(h)}$ is the $i$-th row of $W_0^{(h)}$. Then, $\boldsymbol{x}_j^{(h)}(\boldsymbol{W}_0)$ is $\mathcal{F}_h$-measurable, each summand in the above display is bounded by $a_\sigma \sigma_0^2$ almost surely, and

$$\mathbb{E}[\|\boldsymbol{x}_j^{(h)}\|_2^2 | \mathcal{F}_{h-1}] = a_\sigma \mathbb{E}[\sigma^2(\langle W_{0,1}^{(h)}, \boldsymbol{x}_j^{(h-1)}\rangle) | \mathcal{F}_{h-1}].$$

Thus, by Hoeffding's inequality, conditioned on $\mathcal{F}_{h-1}$, we have

$$\max_{j=1,2,\ldots,n} \left| \|\boldsymbol{x}_j^{(h)}\|_2^2 - a_\sigma \mathbb{E}\big[\sigma^2\big(\langle W_{0,1}^{(h)}, \boldsymbol{x}_j^{(h-1)}\rangle\big) | \mathcal{F}_{h-1}\big] \right| \leq a_\sigma \sigma_0^2 \sqrt{\frac{\log(2n/\delta)}{2m}}, \tag{56}$$

with probability at least $1 - \delta$. Since $\mathbb{E}_{u_0 \sim \mathcal{N}(0,\boldsymbol{I})}[\sigma^2(u_0^\top x)] = \mathbb{E}_{z \sim \mathcal{N}(0,1)}[\sigma^2(z\|x\|_2)]$, we have

$$\mathbb{E}\big[\sigma^2\big(\langle W_{0,1}^{(h)}, \boldsymbol{x}_j^{(h-1)}\rangle\big) | \mathcal{F}_{h-1}\big] = \mathbb{E}_{z \sim \mathcal{N}(0,1)}\big[\sigma^2\big(z \cdot \|\boldsymbol{x}_j^{(h-1)}\|_2\big) | \mathcal{F}_{h-1}\big]. \tag{57}$$

Using $\sigma_0$-boundedness and $\sigma_1$-Lipschitz continuity of $\sigma$, for any $x \in \mathbb{R}^m$,

$$\left| \mathbb{E}[\sigma^2(z)] - \mathbb{E}[\sigma^2(z \cdot \|x\|_2)] \right| \leq 2\sigma_0 \sigma_1 \big| 1 - \|x\|_2 \big|.$$

Hence, dividing both sides by $a_\sigma$, we obtain

$$\left| 1 - a_\sigma \mathbb{E}[\sigma^2(z \cdot \|\boldsymbol{x}_j^{(h-1)}\|_2)] \right| \leq C \big| 1 - \|\boldsymbol{x}_j^{(h-1)}\|_2 \big|. \tag{58}$$

Using (56), (57) and (58), we obtain

$$\left| 1 - \|\boldsymbol{x}_j^{(h)}\|_2 \right| \leq \left| 1 - \|\boldsymbol{x}_j^{(h)}\|_2^2 \right| \leq C \big| 1 - \|\boldsymbol{x}_j^{(h-1)}\|_2 \big| + a_\sigma \sigma_0^2 \sqrt{\frac{\log(2n/\delta)}{2m}}.$$

With the choice

$$m \geq \frac{a_\sigma^2 \sigma_0^4 C^2 \log(2nH/\delta)}{2k^2},$$

the last term above is bounded by $\frac{k}{C}$ and we obtain the recursion

$$\left| 1 - \|\boldsymbol{x}_j^{(h)}\|_2 \right| \leq C \big| 1 - \|\boldsymbol{x}_j^{(h-1)}\|_2 \big| + \frac{k}{C},$$

for all $j \in [n]$ and $h \in [H]$ with probability at least $1 - \delta$. With the choice $k = \frac{C(C-1)}{2(C^H-1)}$, this implies

$$\frac{1}{2} \leq \|\boldsymbol{x}_j^{(h)}\|_2 \leq \frac{3}{2}, \quad \forall j \in [n], \forall h \in [H], \tag{59}$$

with probability at least $1 - \delta$ at initialization. The bound on $\max_{h \in [H]} \|W_0^{(h)}\|_2$ follows from sub-Gaussian random matrix inequality (Theorem 4.4.3) in (Vershynin, 2018). Since $\|c_0\|_2 \leq \sqrt{m}$ and $\|c_0\|_4 \leq m^{1/4}$ almost surely at initialization, the result follows from substituting these inequalities into Lemma B.4 in (Du et al., 2019). $\square$

Using Lemma 3.10, the proof of Corollary 3.11 follows from identical steps as Corollary 3.6.

## B. Omitted Proofs from Section 4

*Proof of Lemma 4.2.* We provide the proof for completeness. Note that $\lambda_{\min}(\mathrm{D}^\top f(w)\mathrm{D}f(w)) = \min_{v \in \mathbb{S}^{p-1}} \|\mathrm{D}f(w)v\|_2^2$. Fix $v \in \mathbb{S}^{p-1}$. Then,

$$
\begin{aligned}
\|\mathrm{D}f(w)v\|_2 &\geq \|\mathrm{D}f(w_0)v\|_2 - \|(\mathrm{D}f(w_0) - \mathrm{D}f(w))v\|_2 \\
&\geq \|\mathrm{D}f(w_0)v\|_2 - \|\mathrm{D}f(w_0) - \mathrm{D}f(w)\| \\
&\geq \|\mathrm{D}f(w_0)v\|_2 - L\|w_0 - w\|_2,
\end{aligned}
$$

where the last inequality follows from the $L$-Lipschitz continuity of $w \mapsto \mathrm{D}f(w)$. Taking minimum over $v \in \mathbb{S}^{p-1}$ on both sides, we conclude that

$$
\sqrt{\lambda_{\min}(\mathrm{D}^\top f(w)\mathrm{D}f(w))} \geq \sqrt{\lambda_{\min}(\mathrm{D}^\top f(w_0)\mathrm{D}f(w_0))} - L\|w - w_0\|_2.
$$

If $\|w - w_0\|_2 \leq \lambda_0/L$, then $\lambda_{\min}(\mathrm{D}^\top f(w)\mathrm{D}f(w)) \geq \lambda_0^2$. $\qquad\square$

*Proof of Lemma 4.5.* Consider two arbitrary points $w, w' \in B$, and let

$$
\gamma(t) := tw + (1 - t)w', \ t \in [0, 1].
$$

Then, we have $\frac{\mathrm{d}f(\gamma(t))}{\mathrm{d}t} = \mathrm{D}f(\gamma(t))(w - w')$ for any $t \in [0, 1]$, which implies that

$$
\begin{aligned}
f(w) - f(w') &= \int_0^1 \mathrm{D}f(\gamma(s))(w - w')\mathrm{d}s \\
&= \mathrm{D}f(w_0)(w - w') + \int_0^1 (\mathrm{D}f(\gamma(s)) - \mathrm{D}f(w_0))(w - w')\mathrm{d}s.
\end{aligned}
$$

First, note that

$$
\|\mathrm{D}f(w_0)(w - w')\|_2 = \sqrt{(w - w')\mathrm{D}^\top f(w_0)\mathrm{D}f(w_0)(w - w')} \geq 2\lambda_0\|w - w'\|_2, \tag{60}
$$

by Assumption 4.1. Then, for any $s \in [0, 1]$, $L$-Lipschitz continuity of $w \mapsto \mathrm{D}f(w)$ (where $L = \sigma_2 X \sqrt{n/m}$) implies

$$
\begin{aligned}
\|(\mathrm{D}f(\gamma(s)) - \mathrm{D}f(w_0))(w - w')\|_2 &\leq L\|\gamma(s) - w_0\|_2\|w - w'\|_2 \\
&\leq L(s\|w - w_0\|_2 + (1 - s)\|w' - w_0\|_2)\|w - w'\|_2 \\
&\leq L\,r_0\|w - w'\|_2 \\
&\leq \lambda_0\|w - w'\|_2, \tag{61}
\end{aligned}
$$

since $w', w \in B$. Using (60) and (61), we obtain

$$
\begin{aligned}
\|f(w) - f(w')\|_2 &\geq \|\mathrm{D}f(w_0)(w - w')\|_2 - \int_0^1 \|(\mathrm{D}f(\gamma(s)) - \mathrm{D}f(w_0))(w - w')\|_2\,\mathrm{d}s \\
&\geq \lambda_0\|w - w'\|_2. \tag{62}
\end{aligned}
$$

Hence, if $w \neq w'$, then $f(w) \neq f(w')$, which implies that $w \mapsto f(w)$ is injective on $B$. Since $\alpha > 0$, injectivity of $f$ on $B$ implies injectivity of $\alpha f$ on $B$. Recall that $\alpha f|_B$ is a smooth immersion on $B$ by Lemma 4.2. Note that $B \subset \mathbb{R}^p$ is a closed ball in $\mathbb{R}^p$, which implies that it is a compact smooth manifold with boundary. Since $\alpha f|_B$ is an injective smooth immersion and $B$ is a compact smooth manifold with boundary, Proposition 4.22 in (Lee, 2012) implies that $\alpha f|_B$ is a smooth embedding. $\qquad\square$

The following lemma, which characterizes the magnitude of the second fundamental form and quantifies curvature, will be fundamental throughout the analysis.

**Lemma B.1** (Curvature bounds). *Take an arbitrary $\alpha f(w) \in \mathcal{M}$ and $u \in \mathcal{T}_{\alpha f(w)}\mathcal{M}\backslash\{0\}$. Let $\beta : [0,1] \to \mathcal{M}$ be any smooth curve such that $\beta_0 = \alpha f(w)$ and $\frac{d\beta_t}{dt}\big|_{t=0} = u$. Since $\alpha f$ is a smooth embedding and an injective immersion, we have a smooth curve $\gamma : [0,1] \to B$ such that $\beta_t := \alpha f(\gamma_t) \in \mathcal{M}$ with $\gamma_0 = w$ and $\frac{d\alpha f(\gamma_t)}{dt}\big|_{t=0} = u$. Then, we have*

$$\left\|\frac{d}{dt}\Big[\boldsymbol{P}(\alpha f(\gamma_t))\Big]\Big|_{t=0}\right\| \leq \frac{2L\|u\|_2}{\alpha \lambda_0^2}. \tag{63}$$

*Proof of Lemma B.1.* In the following, we will establish an upper bound on $\left\|\big[\frac{d}{dt}\boldsymbol{P}(\alpha f(\gamma_t))\big]\big|_{t=0}\right\|$. The first inequality follows from a classical result in perturbation theory.

**Claim.** (Theorem 3.9 in (Stewart & Sun, 1990)) Let $J_i \in \mathbb{R}^{n\times p}$, $i = 1, 2$, be two matrices such that

$$\min\{\lambda_{\min}(J_1^\top J_1), \lambda_{\min}(J_2^\top J_2)\} \geq \lambda^2.$$

Let $P_i = J_i[J_i^\top J_i]^{-1}J_i^\top$, $i = 1, 2$. Then, we have

$$\|P_1 - P_2\| \leq \frac{2}{\lambda}\|J_1 - J_2\|. \tag{64}$$

Recall the definition $\boldsymbol{P}(\alpha f(w)) := \mathrm{D}f(w)\Big[\mathrm{D}^\top f(w)\mathrm{D}f(w)\Big]^{-1}\mathrm{D}^\top f(w)$. For any $t \in [0,1]$, we have $\gamma_t \in B$, thus $\lambda_{\min}\big(\mathrm{D}^\top f(\gamma_t)\mathrm{D}f(\gamma_t)\big) \geq \lambda_0^2$ by Lemma 4.2. Thus, for any $t \in [0,1]$, we have

$$\|\boldsymbol{P}(\alpha f(\gamma_t)) - \boldsymbol{P}(\alpha f(\gamma_0))\| \leq \frac{2}{\lambda_0}\|\mathrm{D}f(\gamma_t) - \mathrm{D}f(\gamma_0)\|.$$

Recall that $w \mapsto \mathrm{D}f(w)$ is globally $L$-Lipschitz where $L = \sigma_2 X\sqrt{n/m}$. Therefore,

$$\|\boldsymbol{P}(\alpha f(\gamma_t)) - \boldsymbol{P}(\alpha f(\gamma_0))\| \leq \frac{2L}{\lambda_0}\|\gamma_t - \gamma_0\|. \tag{65}$$

By using the lower bound in (62), we have

$$\|\gamma_t - \gamma_0\|_2 \leq \frac{1}{\alpha\lambda_0}\|\alpha f(\gamma_t) - \alpha f(\gamma_0)\|.$$

Substituting this into (65), we obtain

$$\|\boldsymbol{P}(\alpha f(\gamma_t)) - \boldsymbol{P}(\alpha f(\gamma_0))\| \leq \frac{2L}{\alpha\lambda_0^2}\|\alpha f(\gamma_t) - \alpha f(\gamma_0)\|.$$

Therefore, we have

$$\left\|\frac{d}{dt}\Big[\boldsymbol{P}(\alpha f(\gamma_t))\Big]\Big|_{t=0}\right\| \leq \frac{2L}{\alpha\lambda_0^2}\cdot\left\|\frac{d\alpha f(\gamma_t)}{dt}\Big|_{t=0}\right\|_2 = \frac{2L\|u\|_2}{\alpha\lambda_0^2} \tag{66}$$

since $\frac{d\alpha f(\gamma_t)}{dt}\big|_{t=0} = u$. $\qquad\square$

*Proof of Lemma 4.9.* Note that $B \subset \mathbb{R}^r$ is compact and $\alpha f|_B$ is continuous, therefore $\mathcal{M}$ is compact. Hence, together with the Riemannian distance $d_\mathcal{M}$, $(\mathcal{M}, d_\mathcal{M})$ is a complete and locally compact length space. Since $B$ is path connected and $\alpha f|_B$ is smooth, $\mathcal{M}$ is path connected, hence there exists a curve of finite length connecting any two points in $\mathcal{M}$. Therefore, by Theorem 2.5.23 in Burago et al. (2001), any two points $\alpha f(w), \alpha f(w') \in \mathcal{M}$ are joined by a minimizing geodesic in $\mathcal{M}$. $\qquad\square$

*Proof of Theorem 4.10.* (a) Fix $\alpha f(w) \in \mathcal{M}$ and $u \in \mathcal{T}_{\alpha f(w)}\mathcal{M}\backslash\{0\}$. Let $\beta : [0,1] \to \mathcal{M}$ be any smooth curve such that $\beta_0 = \alpha f(w)$ and $\frac{d\beta_t}{dt}\big|_{t=0} = u$. Since $\alpha f$ is a smooth embedding by Lemma 4.5, it is a diffeomorphism onto its image $\mathcal{M}$ with smooth inverse. Thus, we have a smooth curve $\gamma : [0,1] \to B$ such that $\beta_t := \alpha f(\gamma_t) \in \mathcal{M}$ with $\gamma_0 = w$ and $\frac{d\alpha f(\gamma_t)}{dt}\big|_{t=0} = u$. Then, the quadratic form for the Riemannian Hessian is

$$\langle u, \text{Hess } g(\alpha f(w))[u]\rangle_{\alpha f(w)}^\mathcal{M} = u^\top \lim_{t\to 0}\frac{\boldsymbol{P}(\alpha f(w))\Big[\text{grad}_{\alpha f(\gamma_t)}^\mathcal{M}g(\alpha f(\gamma_t)) - \text{grad}_{\alpha f(w)}^\mathcal{M}g(\alpha f(w))\Big]}{t} \tag{67}$$

by (5.19) in (Boumal, 2023). Recall that

$$\mathrm{grad}^{\mathcal{M}}_{\alpha f(\gamma_t)} g(\alpha f(\gamma_t)) = \boldsymbol{P}(\alpha f(\gamma_t)) \nabla g(\alpha f(\gamma_t)).$$

Thus, we can make the following decomposition for any $t \in [0,1]$:

$$
\begin{aligned}
u^\top \boldsymbol{P}(\alpha f(w)) & \Big[ \mathrm{grad}^{\mathcal{M}}_{\alpha f(\gamma_t)} g(\alpha f(\gamma_t)) - \mathrm{grad}^{\mathcal{M}}_{\alpha f(w)} g(\alpha f(w)) \Big] \\
&= u^\top \boldsymbol{P}(\alpha f(w)) \Big[ \nabla g(\alpha f(\gamma_t)) - \nabla g(\alpha f(w)) \Big] + u^\top \Big[ \boldsymbol{P}(\alpha f(\gamma_t)) - \boldsymbol{P}(\alpha f(w)) \Big] \nabla g(\alpha f(\gamma_t)).
\end{aligned}
\tag{68}
$$

The first term can be lower bounded as

$$
\begin{aligned}
u^\top \lim_{t \downarrow 0} \frac{\boldsymbol{P}(\alpha f(w)) \Big[ \nabla g(\alpha f(\gamma_t)) - \nabla g(\alpha f(w)) \Big]}{t} &= u^\top \lim_{t \downarrow 0} \frac{\nabla g(\alpha f(\gamma_t)) - \nabla g(\alpha f(\gamma_0))}{t} \\
&= u^\top \nabla^2 g(\alpha f(\gamma_0)) \frac{\mathrm{d}\alpha f(\gamma_t)}{\mathrm{d}t}\Big|_{t=0} \\
&\geq \nu \|u\|_2^2 = \nu (\|u\|^{\mathcal{M}}_{\alpha f(w)})^2,
\end{aligned}
\tag{69}
$$

since $h \mapsto g(h)$ is (Euclidean) $\nu$-strongly convex, thus $\nabla^2 g(h) \succeq \nu I$ for all $h \in \mathbb{R}^n$, and $u \in \mathcal{T}_{\alpha f(w)} \mathcal{M}$, thus $\boldsymbol{P}(\alpha f(w)) u = u$, and $\boldsymbol{P}(\alpha f(w))$ is symmetric.

For the second term, let

$$\mathrm{Lip}^{\mathcal{C}}_g := \sup_{z \in \mathcal{C}} \|\nabla g(z)\|,$$

for any $\mathcal{C} \subset \mathbb{R}^n$. Then, since $\alpha f(\gamma_t) \in \mathcal{M}$ for all $t \in [0,1]$, we have

$$
\begin{aligned}
u^\top \lim_{t \downarrow 0} \frac{[\boldsymbol{P}(\alpha f(\gamma_t)) - \boldsymbol{P}(\alpha f(w))] \nabla g(\alpha f(\gamma_t))}{t} &\geq -\mathrm{Lip}^{\mathcal{M}}_g \cdot \|u\|_2 \cdot \lim_{t \downarrow 0} \frac{\|\boldsymbol{P}(\alpha f(\gamma_t)) - \boldsymbol{P}(\alpha f(w))\|}{t} \\
&= -\mathrm{Lip}^{\mathcal{M}}_g \cdot \|u\|_2 \cdot \Big\| \frac{\mathrm{d}}{\mathrm{d}t} \Big[ \boldsymbol{P}(\alpha f(\gamma_t)) \Big]\Big|_{t=0} \Big\|.
\end{aligned}
\tag{70}
$$

In order to bound the last term above, we use Lemma B.1. Substituting (66) into (70), we obtain

$$
u^\top \lim_{t \downarrow 0} \frac{[\boldsymbol{P}(\alpha f(\gamma_t)) - \boldsymbol{P}(\alpha f(w))] \nabla g(\alpha f(\gamma_t))}{t} \geq -\frac{2L \cdot \mathrm{Lip}^{\mathcal{M}}_g}{\alpha \lambda_0^2} \cdot \|u\|_2^2.
\tag{71}
$$

Substituting (69) and (71) into (67) by using the decomposition (68), we conclude that

$$
\langle u, \mathrm{Hess}\, g(\alpha f(w))[u] \rangle^{\mathcal{M}}_{\alpha f(w)} \geq \nu \|u\|_2^2 - \frac{2L \cdot \mathrm{Lip}^{\mathcal{M}}_g}{\alpha \lambda_0^2} \cdot \|u\|_2^2.
\tag{72}
$$

Finally, for any $w \in B$, we have

$$
\begin{aligned}
\|\nabla g(\alpha f(w))\|_2 &= \|\nabla g(\alpha f(w)) - \nabla g(f^\star)\|_2 \\
&\leq \mu \alpha \|f(w) - f(w_0)\|_2 + \mu \|f^\star\|_2 \\
&\leq \mu \alpha \mathrm{Lip}_f \|w - w_0\|_2 + \mu \|f^\star\|_2 \leq \mu \Big( \alpha \mathrm{Lip}_f r_0 + \|f^\star\|_2 \Big),
\end{aligned}
$$

which follows from $\mu$-Lipschitz continuity of $\nabla g$, $\nabla g(f^\star) = 0$, and $f(w_0) = 0$. Hence, we have

$$
\mathrm{Lip}^{\mathcal{M}}_g \leq \mu \Big( \alpha \mathrm{Lip}_f r_0 + \|f^\star\| \Big).
\tag{73}
$$

By choosing $\alpha$ as stated in the theorem, we ensure that $\frac{2L\mu\|f^\star\|}{\alpha\lambda_0^2} \leq \frac{\nu}{2}$. Since $r_0 \leq \frac{\nu\lambda_0^2}{4\mu L \mathrm{Lip}_f}$, we also have $\frac{2L\mu\mathrm{Lip}_f r_0}{\lambda_0^2} \leq \frac{\nu}{2}$. Using these two results, (72) implies the positive semi-definiteness of the Riemannian Hessian on $\mathcal{M}$, which implies geodesic convexity of $g|_{\mathcal{M}}$ since $\mathcal{M}$ is a geodesically convex set.

(b) $g|_{\mathcal{M}} : \mathcal{M} \to \mathbb{R}$ is a geodesically convex function, and $\mathcal{S} := \{z \in \mathcal{M} : g(z) \leq g(0)\}$ is a sublevel set for $g$. Then, $\mathcal{S}$ is geodesically convex by Proposition 11.8 in (Boumal, 2023).

(c) Fix $\alpha f(w) \in \mathcal{S}$ and $u \in \mathcal{T}_{\alpha f(w)}\mathcal{M}\backslash\{0\}$. Let $\beta : [0,1] \to \mathcal{S}$ be any smooth curve such that $\beta_0 = \alpha f(w)$ and $\frac{d\beta_t}{dt}\big|_{t=0} = u$, and let $\gamma : [0,1] \to B$ such that $\beta_t := \alpha f(\gamma_t) \in \mathcal{S}$ with $\gamma_0 = w$ and $\frac{d\alpha f(\gamma_t)}{dt}\big|_{t=0} = u$. The proof of geodesic strong convexity of $g$ in $\mathcal{S}$ follows identical steps as (a) until (72), where $\mathrm{Lip}_g^{\mathcal{S}}$ replaces $\mathrm{Lip}_g^{\mathcal{M}}$ since $\alpha f(\gamma_t) \in \mathcal{S}$ for all $t \in [0,1]$. Now, note that

$$\|\nabla g(\alpha f(w))\| \leq \mu\|\alpha f(w) - f^{\star}\| \leq \mu\sqrt{\frac{2(g(\alpha f(w)) - g(f^{\star}))}{\nu}}$$

$$\leq \mu\sqrt{\frac{2g(0)}{\nu}}$$

for any $\alpha f(w) \in \mathcal{S}$ (i.e., $g(\alpha f(w)) \leq g(0)$). This implies that $\mathrm{Lip}_g^{\mathcal{S}} \leq \mu\sqrt{\frac{2g(0)}{\nu}}$.

(d) Similar to the proof of (c), fix $\alpha f(w) \in \mathcal{S}$ and $u \in \mathcal{T}_{\alpha f(w)}\mathcal{M}\backslash\{0\}$. Let $\beta : [0,1] \to \mathcal{S}$ be any smooth curve such that $\beta_0 = \alpha f(w)$ and $\frac{d\beta_t}{dt}\big|_{t=0} = u$, and let $\gamma : [0,1] \to B$ such that $\beta_t := \alpha f(\gamma_t) \in \mathcal{S}$ with $\gamma_0 = w$ and $\frac{d\alpha f(\gamma_t)}{dt}\big|_{t=0} = u$. We aim to find an upper bound for the quadratic form in (67) using the decomposition in (68). Similar to (69), we have

$$u^{\top}\lim_{t\downarrow 0}\frac{\boldsymbol{P}(\alpha f(w))\Big[\nabla g(\alpha f(\gamma_t)) - \nabla g(\alpha f(w))\Big]}{t} = u^{\top}\nabla^2 g(\alpha f(w))\frac{d\alpha f(\gamma_t)}{dt}\Big|_{t=0}$$

$$= u^{\top}\nabla^2 g(\alpha f(w))u$$

$$\leq \mu\|u\|_2^2, \tag{74}$$

where the second line holds since $\frac{d\alpha f(\gamma_t)}{dt}\big|_{t=0} = u$ and the inequality is due to $\sup_{h\in\mathbb{R}^n}\|\nabla^2 g(h)\| \leq \mu$ from the Lipschitz continuity of $\nabla g$. We also have

$$u^{\top}\lim_{t\downarrow 0}\frac{[\boldsymbol{P}(\alpha f(\gamma_t)) - \boldsymbol{P}(\alpha f(w))]\nabla g(\alpha f(\gamma_t))}{t} \leq \mathrm{Lip}_g^{\mathcal{S}} \cdot \|u\|_2 \cdot \lim_{t\downarrow 0}\frac{\|\boldsymbol{P}(\alpha f(\gamma_t)) - \boldsymbol{P}(\alpha f(w))\|}{t}$$

$$= \mathrm{Lip}_g^{\mathcal{S}} \cdot \|u\|_2 \cdot \left\|\frac{d}{dt}\Big[\boldsymbol{P}(\alpha f(\gamma_t))\Big]\Big|_{t=0}\right\|. \tag{75}$$

Substituting (66) into the above inequality, we obtain

$$u^{\top}\lim_{t\downarrow 0}\frac{[\boldsymbol{P}(\alpha f(\gamma_t)) - \boldsymbol{P}(\alpha f(w))]\nabla g(\alpha f(\gamma_t))}{t} \leq \frac{2L \cdot \mathrm{Lip}_g^{\mathcal{S}}}{\alpha\lambda_0^2} \cdot \|u\|_2^2 \leq \frac{\nu}{2}\|u\|_2^2$$

$$\leq \frac{\mu}{2}\|u\|_2^2, \tag{76}$$

where the last inequality holds since $\nu I \preccurlyeq \nabla^2 g(h) \preccurlyeq \mu I$ for all $h \in \mathbb{R}^n$. From (74) and (76), the decomposition in (68) implies

$$\langle u, \mathrm{Hess}\, g(\alpha f(w))[u]\rangle_{\alpha f(w)}^{\mathcal{M}} \leq \frac{3\mu}{2}\|u\|_2^2,$$

which concludes the proof.

$\square$

*Proof of Lemma 4.13.* Let $w^{\star} := w_{\alpha}^{\star}$ for notational simplicity. Since $\alpha f(w^{\star}), \alpha f(w_t) \in \mathcal{S}$ for $t < T$ and $\mathcal{S}$ is geodesically convex, there exists $v_t \in T_{\alpha f(w^{\star})}\mathcal{M}$ such that $\mathrm{Exp}_{\alpha f(w^{\star})}(v_t) = \alpha f(w_t)$ and the geodesic $c(\xi) = \mathrm{Exp}_{\alpha f(w^{\star})}(\xi v_t)$ lies in $\mathcal{S}$. In particular, $c(0) = \alpha f(w^{\star}), c(1) = \alpha f(w_t)$. For any $(\alpha f(w), u)$ in the tangent bundle, let $\gamma(\xi) := \mathrm{Exp}_{\alpha f(w)}(\xi u)$ be corresponding geodesic. Then, let $P_{\xi u} := \mathrm{PT}_{\xi\leftarrow 0}^{\gamma}$ be the parallel transport from $\alpha f(w)$ to $\mathrm{Exp}_{\alpha f(w)}(\xi u)$ along $\gamma$. In particular, along $c(\xi) = \mathrm{Exp}_{\alpha f(w^{\star})}(\xi v_t)$, let $P_{v_t} := \mathrm{PT}_{1\leftarrow 0}^c$. $P_{v_t}^{-1}$ is an isometry (Lee, 2018). Since $g|_{\mathcal{S}}$ has geodesically $\frac{3\mu}{2}$-Lipschitz continuous gradients by Theorem 4.10(d), we have

$$\|P_{v_t}^{-1}\mathrm{grad}_{\alpha f(w_t)}^{\mathcal{M}}g(\alpha f(w_t)) - \mathrm{grad}_{\alpha f(w^{\star})}^{\mathcal{M}}g(\alpha f(w^{\star}))\| \leq \frac{3\mu}{2}\|v_t\|$$

by Prop. 10.53 in (Boumal, 2023). Since $\mathrm{grad}^{\mathcal{M}}_{\alpha f(w^\star)} g(\alpha f(w^\star)) = 0$ and $P^{-1}_{v_t}$ is an isometry, we have

$$\|\mathrm{grad}^{\mathcal{M}}_{\alpha f(w_t)} g(\alpha f(w_t))\| \leq \frac{3\mu}{2}\|v_t\|.$$

Theorem 4.10 implies that $g|_{\mathcal{S}}$ is $\frac{\nu}{2}$-geodesically strongly convex, therefore

$$\frac{\nu}{4}\|v_t\|^2 \leq g(\alpha f(w_t)) - g(\alpha f(w^\star))$$

$$\leq \frac{1}{\nu}\|\mathrm{grad}^{\mathcal{M}}_{\alpha f(w_t)} g(\alpha f(w_t))\|^2_{\alpha f(w_t)}.$$

where the second inequality follows from the Riemannian Polyak-Lojasiewicz inequality (Boumal, 2023). □

*Proof of Theorem 4.14.* From Lemma 4.3, recall that we have

$$\frac{\mathrm{d}V_t}{\mathrm{d}t} = -\|\mathrm{grad}^{\mathcal{M}}_{\alpha f(w_t)} g(\alpha f(w_t))\|^2_2 \text{ for any } t \in (0, T).$$

For $\alpha \geq \alpha_0$, $g|_{\mathcal{S}}$ is $\nu/2$ geodesically strong convex by Theorem 4.10. Hence, the Riemannian Polyak-Lojasiewicz inequality in Lemma 4.13 implies that

$$\dot{V}_t \leq -\nu V_t.$$

Thus, Grönwall's lemma implies

$$V_t \leq V_0 \exp(-\nu t) \text{ for any } t \in [0, T). \tag{77}$$

To show that $T = \infty$, take $t \in [0, T)$. Then,

$$\|w_t - w_0\|_2 \leq \int_0^t \|\dot{w}_s\|\mathrm{d}s$$

$$= \frac{1}{\alpha} \int_0^t \|\nabla g(\alpha f(w_s))\|_{\boldsymbol{A}_s}\mathrm{d}s, \tag{78}$$

where

$$\boldsymbol{A}_s := \mathrm{D}f(w_s)\boldsymbol{H}^{-2}(w_s)\mathrm{D}^\top f(w_s) \text{ for } s < T.$$

Since $s < t < T$, we have $w_s \in B$, thus $\boldsymbol{H}(w_s) \succcurlyeq \lambda_0^2 \boldsymbol{I}$. This implies that

$$\|\nabla g(\alpha f(w_s))\|^2_{\boldsymbol{A}_s} \leq \frac{1}{\lambda_0^2}\|\nabla g(\alpha f(w_s))\|^2_{\boldsymbol{P}(\alpha f(w_s))}$$

$$= \frac{1}{\lambda_0^2}\|\boldsymbol{P}(\alpha f(w_s))\nabla g(\alpha f(w_s))\|^2_2$$

$$= \frac{1}{\lambda_0^2}\|\mathrm{grad}^{\mathcal{M}}_{\alpha f(w_s)} g(\alpha f(w_s))\|^2_2. \tag{79}$$

By using (22) in Lemma 4.13, we have

$$\|\mathrm{grad}^{\mathcal{M}}_{\alpha f(w_s)} g(\alpha f(w_s))\|^2_{\alpha f(w_s)} \leq \frac{9\mu^2}{4}\|v_s\|^2_2 \leq \frac{9\mu^2}{\nu}V_s.$$

Using the error bound (77), we obtain

$$\|\mathrm{grad}^{\mathcal{M}}_{\alpha f(w_s)} g(\alpha f(w_s))\|_{\alpha f(w_s)} \leq \frac{3\mu\sqrt{V_s}}{\sqrt{\nu}} \leq \frac{3\mu\sqrt{V_0}}{\sqrt{\nu}}e^{-\nu s/2} \leq \frac{3\mu\sqrt{g(0)}}{\sqrt{\nu}}e^{-\nu s/2} \text{ for any } s < T. \tag{80}$$

Substituting (79) and (80) into (78), we obtain

$$\|w_t - w_0\| \leq \frac{3\mu\sqrt{g(0)}}{\alpha\lambda_0\sqrt{\nu}} \int_0^t \exp(-\nu s/2)\mathrm{d}s \leq \frac{6\mu\sqrt{g(0)}}{\alpha\lambda_0\sqrt{\nu^3}}.$$

Hence, $\alpha \geq \frac{6\mu\sqrt{g(0)}}{\lambda_0 r_0 \nu^{\frac{3}{2}}}$ yields $\|w_t - w_0\| \leq r_0$ and $\mathrm{D}^\top f(w_t)\mathrm{D}f(w_t) \succcurlyeq \lambda_0^2 \boldsymbol{I}$. Hence, $T = \infty$. □

## C. Implicit Bias of Gauss–Newton for Quadratic Loss

We characterize the implicit bias of the linearized (lazy) model trained under the Gauss–Newton dynamics. In the quadratic loss setting, the induced flow converges to the minimum-norm interpolant in the metric defined by the Gauss–Newton preconditioner, i.e., the solution of a constrained least-norm problem with norm $\| \cdot \|_{\boldsymbol{H}_\rho(w_0)}$. This is the natural analogue of the classical "minimum Euclidean norm" bias of gradient flow for linear regression, extended here to a nontrivial preconditioning geometry. Our result builds on the implicit bias argument for linear least squares (Bach, 2024) and connects to the lazy-training/linearization perspective of overparameterized networks (Chizat et al., 2019).

**Proposition C.1** (Implicit bias of Gauss–Newton for quadratic loss). *Under Assumption 3.1, for any $u \in \mathbb{R}^p$, let*

$$\bar{f}_0(u) = \mathrm{D}f(w_0)(u - w_0)$$

*and consider*

$$\dot{u}_t = -[\boldsymbol{H}_\rho(w_0)]^{-1}\mathrm{D}^\top f(w_0)(\bar{f}_0(u_t) - y), \quad t \geq 0,$$

*with $u_0 = w_0$ for $\rho \in (0, \frac{\lambda^2}{1+\lambda^2}]$. Then, $u_t \to u^\star$ as $t \to \infty$, where $u^\star$ is the solution of*

$$\min_{u \in \mathbb{R}^p} \|u - w_0\|^2_{\boldsymbol{H}_\rho(w_0)} \quad s.t. \quad \bar{f}_0(u) = y.$$

*Proof of Proposition C.1.* Under Assumption 3.1, we have $\mathrm{D}f(w_0)\boldsymbol{H}_\rho^{-1}(w_0)\mathrm{D}^\top f(w_0) \succ 0$, thus $\lim_{t \to \infty} \bar{f}_0(u_t) = y$ using the same Grönwall argument as Theorem 3.4. Let

$$\beta_t = -\int_0^t (y - \bar{f}_0(u_s))\mathrm{d}s, \quad t \geq 0.$$

Then, $u_t - w_0 = [\boldsymbol{H}_\rho(w_0)]^{-1}\mathrm{D}^\top f(w_0)\beta_t, \quad t \geq 0$. Since $\bar{f}_0(u_t) \to y$ as $t \to \infty$,

$$\lim_{t \to \infty} \mathrm{D}f(w_0)[\boldsymbol{H}_\rho(w_0)]^{-1}\mathrm{D}^\top f(w_0)\beta_t = y \quad \Rightarrow \quad \lim_{t \to \infty} \beta_t = (\mathrm{D}f(w_0)[\boldsymbol{H}_\rho(w_0)]^{-1}\mathrm{D}^\top f(w_0))^{-1}y.$$

Therefore,

$$\lim_{t \to \infty} \{u_t - w_0\} = \boldsymbol{H}_\rho^{-1}(w_0)\mathrm{D}^\top f(w_0)\Big(\mathrm{D}f(w_0)\boldsymbol{H}_\rho^{-1}(w_0)\mathrm{D}^\top f(w_0)\Big)^{-1}y.$$

By Karush-Kuhn-Tucker (KKT) conditions (Bach, 2024; Boyd & Vandenberghe, 2004), the limit yields $u^\star - w_0$ with minimum $\boldsymbol{H}_\rho(w_0)$-norm subject to the affine constraints $\mathrm{D}f(w_0)(u - w_0) = y$. $\square$

## D. Numerical Experiments

This appendix reports small-scale experiments on the empirical risk under Gauss—Newton dynamics in both over- and underparameterized regression settings with gradient flow as a baseline. In both regimes, we use the loss function $g(\psi) = \frac{1}{2}\|\psi - y\|_2^2$ where $y = [y_1, y_2, \ldots, y_n]^\top \in \mathbb{R}^n$.

**Single index model.** We consider a single-index model with a training set $\mathcal{D} = \{(x_j, y_j) \in \mathbb{R}^d \times \mathbb{R} : j = 1, 2, \ldots, n\}$ where the input is $x_j \sim_{\mathrm{iid}} \mathrm{Unif}(\mathbb{S}^{d-1})$ and the label is

$$y_j = \mathrm{ReLU}(u^\top x_j) + \epsilon_j, \tag{81}$$

where $\mathrm{ReLU}(z) = \max\{0, z\}$, $u \in \mathbb{R}^d$ is the fixed target direction and $\epsilon_j \sim \mathcal{N}(0, 1)$ is the noise for $j = 1, 2, \ldots, n$. This input distribution leads to small $\lambda_{\min}(\boldsymbol{K}_0)$ (Karhadkar et al., 2024).

**California Housing dataset.** In the second set of experiments, we consider the California Housing dataset $\mathcal{D} := \{(x_j, y_j) \in \mathbb{R}^d \times \mathbb{R} : j = 1, 2, \ldots, n\}$ (Pace & Barry, 1997), where each feature vector $x_j \in \mathbb{R}^d$ with $d = 8$ represents normalized housing-related attributes, and $y_j \in \mathbb{R}$ represents median house value for $j = 1, 2, \ldots, n$. We form a training set by uniformly subsampling n examples at random.

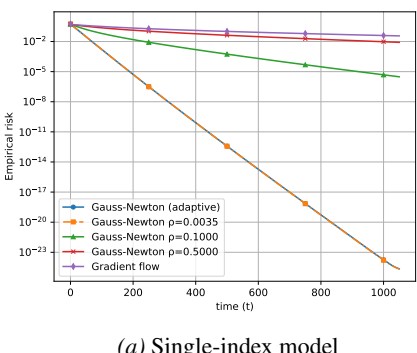 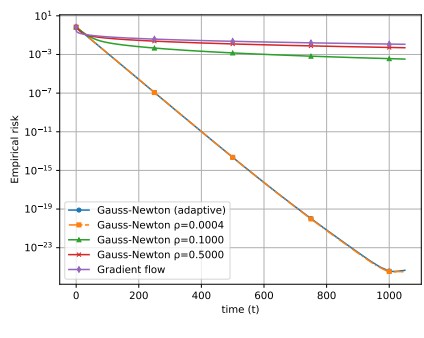

*(a)* Single-index model          *(b)* California Housing

*Figure 2.* $\frac{1}{n} g(\cdot)$ in the overparameterized regime under the Gauss–Newton dynamics with various regularization schemes $\rho = (\rho_t)_{t \geq 0}$. Gradient flow $(\rho_t = 1)$ suffers from slow convergence due to the ill-conditioned neural tangent kernel, while the Gauss–Newton with appropriate constant or adaptive damping schedules achieve fast exponential convergence rates.

## D.1. Overparameterized regime

We consider an overparameterized problem with $p \gg n$ under $\rho^{\mathrm{ada}}$ and various constant $\rho^{\mathrm{con}}$ damping schedules in Figure 2. For the single-index model, we use a dataset of $n = 800$ samples of ambient dimension $d = 16$ and a $\mathtt{tanh}$ neural network with $p = 9600$ parameters. For the California Housing dataset, we randomly subsample a training set of size $n = 800$, and use a $\mathtt{tanh}$ neural network with $p = 6400$ parameters. The parameters are trained by using the Gauss–Newton method with various regularization choices: (i) adaptive damping $\rho_t = \frac{\frac{1}{4}\lambda_t^2}{1 + \frac{1}{4}\lambda_t^2}$, and (ii) constant data-based damping with $\rho_t = \frac{\lambda^2}{\lambda^2 + 1}$ where $\lambda_{\min}(\boldsymbol{K}_0) = 4\lambda^2$. Note that $\rho_t = 1.0$ corresponds to the (non-preconditioned) gradient flow. The continuous-time dynamics are simulated by using Euler's method. $\rho_t = \frac{a\lambda_t^2}{1 + a\lambda_t^2}$ in Lemma 3.3 for $a > 0$ yields $V_t \leq V_0 \exp\left(-2\nu \frac{1 + a\lambda^2}{1 + a} t\right)$.

In these examples, $\lambda_{\min}(\boldsymbol{K}_0) \approx 1.4 \times 10^{-2}$ and $\lambda_{\min}(\boldsymbol{K}_0) \approx 1.6 \times 10^{-3}$, respectively, thus the gradient flow suffers from slow convergence, while Gauss–Newton achieves faster convergence. In particular, the adaptive and the constant damping factors lead to fast exponential convergence in empirical risk (linear on a log-scale), in line with the theoretical results in Theorem 3.4. Note that in the kernel regime with large $\alpha\sqrt{m}$ that we consider, we have $\lambda_t^2 / (1 + \lambda_t^2)$ close to $\lambda^2 / (1 + \lambda^2)$, thus adaptive and data-dependent constant damping choices yield very similar empirical risk performance as predicted.

## D.2. Underparameterized Regime

We investigate the performance of (undamped) Gauss–Newton dynamics and gradient flow in the underparameterized regression problems for the single-index model (81) with $g(\psi) = \frac{1}{2}\|\psi - y\|_2^2$ and $n = 800$ randomly-chosen samples and $p = 256$ parameters. We demonstrate the impact of different $\alpha > 0$ and the impact of Gauss–Newton preconditioning in Figure 3. A larger scaling factor $\alpha$ keeps the trajectory closer to $w_0$, thereby shrinking the admissible parameter set and

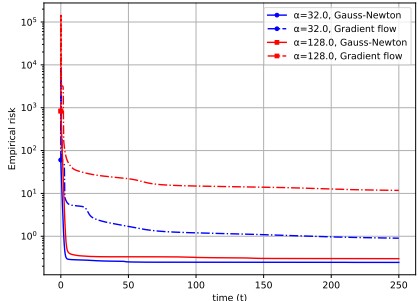

*Figure 3.* Empirical risk $g(\cdot)/n$ in the underparameterized regime under the Gauss–Newton and gradient flow dynamics for various $\alpha$.

changing the best achievable in-class predictor in the underparameterized setting. This trend is consistent with Figure 3: smaller $\alpha$ allows a larger effective search region and typically achieves a lower in-class optimal risk.

