# OpenReview forum: "Geometric Convergence of Gauss–Newton for Neural Networks: Riemannian Geometry and Adaptive Damping"
_ICML.cc/2026/Conference — ICML 2026 regular_

### Official Review · Reviewer_hwsY · 2026-03-10

**Soundness:** 3
**Presentation:** 3
**Significance:** 3
**Originality:** 3
**Overall Recommendation:** 4
**Confidence:** 3

**Summary:**

This paper proposed theoretical studies on the non-asymptotic convergence bounds for the Gauss-Newton method separately in the under- and over-parameterized regimes, respectively through Riemannian gradient flow and survature-aware regularization. The analytics suggested regimes where Gauss-Newton is faster than first-order methods in the near-initialization regime.

**Compliance With Llm Reviewing Policy:**

Affirmed.

**Final Justification:**

I believe the contribution of this paper is worth publishing at this conference, and my concerns are greatly resolved during the rebuttal stage. Therefore, my justifications for this manuscript have not changed, and I suggest final acceptance.

**Key Questions For Authors:**

Q1: One minor suggestion is to move 1.3. Notation to the beginning of 2. Problem Setting for clarity.

Q2: Can you discuss the generality of Assumption 4.1, categorizing the near-initialization regime by the radius $r_0$? While the convergence rate in Theorem 4.14 is independent of the minimum eigenvalue $\lambda_0$, is it possible that the safe zone partially depends on $\lambda$ and degrades the convergence rate softly?

Q3: Regarding Collary 3.11, if the strong convexity modulus $v$ is efficiently small (for a flat landscape), would the convergence bound be ill-conditioned and lead to an unacceptable timestep $t$? Please also consider if $v$ has a soft dependency on the parameters $p$, as deep networks with larger dimensions may induce flatness in the solution landscape.

Q4: Can you discuss how the derived convergence bound compares to the information-theoretic lower bounds for non-linear regression?

**Limitations:**

yes

**Strengths And Weaknesses:**

1. Soundness: The paper's claim is well grounded from theoretical justifications with sound remarks on assumptions and conditions.
2. Presentation: This study is well-presented with a clear architecture and concise languages. The presentation of theoretical results is clean and easy to follow. One of the remaining concerns is the relatively weak motivation from the presentation perspective, though this is understandable.
3. Significance: The theoretical implications of this paper suggested benign properties of the Gauss-Newton method w.r.t. first-order methods from the perspective of Riemannian gradient flow in the under-parameterized regime provide valuable suggestions to understandings of the training dynamics of deep neural networks.
4. Originality: This paper contributes from both a structured convergence analysis for the Gauss-Newton method in under- and over-parameterized regimes, and the introduction of tools from Riemannian optimization, which expand the research scope of the properties of optimization methods.

---

> ### Author Rebuttal · Authors · 2026-03-31
>
> We would like to thank the reviewer for the insightful and constructive feedback. Below, we provide answers to the reviewer's questions.
>
> **Q1.** We thank the reviewer for the suggestion. We agree that this would improve readability, and we will move Section 1.3 (Notation) to the beginning of Section 2 (Problem Setting) in the revision.
>
> **Q2.** Assumption 4.1 is a local regularity/non-degeneracy condition: it requires $Df(w)$ to remain well-conditioned and smooth on the ball $B(w_0,r_0)$ as in (Chizat et al., 2019; Du et al., 2018, 2019). The near-initialization regime is quantified by $r_0$, which is chosen in the proof to ensure these bounds hold.
>
> The role of $\lambda_0$ is through the safe zone: while the rate in Theorem 4.14 is independent of $\lambda_0$, the certified neighborhood size $r_0$ depends on $\lambda_0$. Thus, poor conditioning can shrink the guaranteed region where the assumptions hold, but it does _not_ appear in the exponential convergence rate. We will clarify this separation in the revision.
>
> **Q3.** For Corollary 3.11, a small $\nu$ yields slower convergence since $V_t \leq V_0 \exp(-\frac{2\nu\lambda^2}{\rho+(1-\rho)\lambda^2}t)$. Hence, if $\nu$ is very small, the bound may require a large $t$ to reach a target accuracy, as is also the case for gradient flow (Chizat et al., 2019).
>
> Our analysis in this work indicates that one benefit of generalized Gauss-Newton (GGN) is to mitigate this impact of small $\nu$ locally. In Theorem 3.8, after entering a suitable neighborhood of the optimal parameter and switching to the local damping schedule, the convergence rate of GGN becomes independent of $\nu$ locally (Line 272). Thus, the interpretation is that classical damped GN remains globally sensitive to small $\nu$, whereas GGN is introduced precisely to remove this dependence on $\nu$ locally.
>
> Regarding dependence on the parameter dimension $p$: in our framework, $\nu$ is the strong convexity modulus of the loss $g$ in prediction space ($\mathbb{R}^n$), not parameter space. As such, Corollary 3.11 studies curvature in prediction space through $\nu$, rather than how the neural network parameterization shapes the curvature of the objective in parameter space.
>
> **Q4.** Our convergence bounds are _optimization_ guarantees to obtain a minimizer of the _empirical risk_ under Gauss–Newton dynamics. By contrast, information-theoretic lower bounds for nonlinear regression are typically _statistical_ minimax bounds that quantify the smallest achievable _prediction or estimation error_ over a function class. These two types of results address different bottlenecks and are therefore not directly comparable. A direct comparison would require an additional statistical analysis translating optimization error into prediction or estimation error under a specified nonlinear regression model, which is outside the scope of the present work. A comparison in a statistical setting would indeed be interesting future work.

---

> > ### Author Rebuttal · Reviewer_hwsY · 2026-04-01
> >
> > The author's response to my concerns is satisfactory and greatly appreciated. I will maintain my score.

---

### Official Review · Reviewer_btjJ · 2026-03-11

**Soundness:** 4
**Presentation:** 4
**Significance:** 3
**Originality:** 3
**Overall Recommendation:** 5
**Confidence:** 2

**Summary:**

This paper studies the convergence behavior of the Gauss–Newton method for training neural networks in both overparameterized and underparameterized regimes under the near-initialization setting. In the overparameterized regime, the authors analyze damped GN dynamics and identify adaptive as well as practical constant damping schedules that yield geometric convergence rates robust to ill-conditioned kernel matrices. In particular, their bounds do not deteriorate with the minimum eigenvalue of the NTK, and for generalized GN they further obtain sharper local rates by exploiting the curvature of the loss.
In the underparameterized regime, where standard kernel based arguments break down because the kernel matrix is singular, this paper develops a Riemannian geometric framework. The authors show that the GN flow in parameter space induces a Riemannian gradient flow on a low dimensional embedded submanifold of predictor space. This paper establish geodesic convexity, geodesic strong convexity, and geodesic smoothness of the loss on the relevant manifold, which leads to last iterate geometric convergence to an optimal in class predictor without explicit regularization.

**Compliance With Llm Reviewing Policy:**

Affirmed.

**Final Justification:**

This paper presents important conclusions, is clearly written, and provides rigorous proofs with strong technical depth. I am inclined to recommend acceptance.

**Key Questions For Authors:**

1. Could clarify to what extent the proposed analysis and conclusions extend beyond the NTK/lazy training regime?

2. Could elaborate on the practical implications of these results for designing optimization algorithms in real neural network training?

**Limitations:**

yes

**Strengths And Weaknesses:**

**Strengths**

**Soundness**:  This paper provides rigorous theoretical analysis of Gauss–Newton dynamics in both overparameterized and underparameterized regimes, with clearly stated assumptions and non-asymptotic convergence guarantees. The proofs are structured and supported by a detailed appendix. The use of adaptive damping and Riemannian optimization tools is mathematically well motivated and appropriate for addressing conditioning issues and kernel degeneracy.

**Presentation**: The paper is generally well written and logically organized. The progression from the overparameterized analysis to the Riemannian framework in the underparameterized regime is clear and helps isolate the main ideas. Definitions, assumptions, and results are stated precisely, making the narrative relatively easy to follow for readers familiar with optimization theory.

**Significance**: Understanding the optimization dynamics of second order methods such as Gauss–Newton in neural network training is an important problem. The results suggest that Gauss–Newton can avoid conditioning bottlenecks that slow down first order methods and achieve geometric convergence under realistic settings. These insights could influence future theoretical studies of preconditioned optimization and second order methods in deep learning.

**Originality**: The work provides several original perspectives. In particular, interpreting Gauss–Newton dynamics in the underparameterized regime as a Riemannian gradient flow on a predictor space submanifold is a novel conceptual contribution. The analysis of adaptive damping schedules that yield conditioning robust convergence rates also offers new theoretical insights beyond existing NTK analyses.

---

**Weaknesses**

**Soundness**: The near-initialization regime may limit the applicability of the theory to practical deep learning settings.

**Significance**: While the theoretical insights are valuable, the practical implications for real-world neural network training remain somewhat indirect, as empirical validation or comparisons with modern second-order methods are limited.

---

> ### Author Rebuttal · Authors · 2026-03-31
>
> We would like to thank the reviewer for the insightful and constructive feedback. Below, we provide our answers to the reviewer's questions.
>
> **Beyond the NTK regime.** Our results are proved under non-degeneracy conditions, and a sufficient condition to ensure this is initial non-degeneracy and near-initialization dynamics (Chizat et al., 2019). The Riemannian analysis in the underparameterized regime holds as long as $Df(w_t)$ is rank-$p$ so that the predictor manifold and its induced geometry are well-defined along the path. In the paper, we ensured this by devising a near-initialization analysis, and we believe that this analysis can be established beyond this regime as long as the stated geometric assumptions hold. In the overparameterized regime, the design principle for the adaptive damping is driven by the energy dissipation inequality (EDI) in Lemma 3.3, which can be adapted to control the evolution in predictor space beyond the lazy training regime. Extending the analysis to feature learning regimes (e.g., mean-field regime), and connecting our conditions to non-NTK convergence criteria (e.g., local PL-type conditions; Chatterjee, 2022) are interesting directions for future work.
>
> **Practical implications.** We thank the reviewer for this question. Our results suggest two concrete algorithmic lessons for training.
> - _Curvature-aware damping matters:_ In the overparameterized regime, we identify Levenberg–Marquardt damping schedules that yield provably fast convergence. This motivates choosing damping as a function of local curvature and data geometry (as in our $\rho_t$ rule) to regularize potentially rank-deficient curvature surrogates, such as the (generalized) Gauss-Newton matrix. As an example, Theorem 3.8 proves that using a two-phase damping schedule yields fast convergence rates for Gauss-Newton. Moreover, Corollary 3.6 suggests $\rho_t$ does not have to be updated at every step in certain regimes, which can reduce the computational overhead of evaluating $\rho_t$ at every iteration.
> - _Benefits of Gauss-Newton preconditioning:_ Our analysis identifies regimes where the convergence rate of Gauss-Newton does not deteriorate with poor kernel/Gram conditioning, unlike first-order methods, suggesting that geometry-aware preconditioned updates can be particularly beneficial on ill-conditioned problems. In addition, our local analysis of generalized Gauss–Newton shows that, after entering a suitable neighborhood, the convergence rate becomes independent of $\nu$. In practice, this supports the use of (approximate) Gauss-Newton with adaptive damping, such as the $\rho_t$ choices considered here.
>
> **References**
>
> - S. Chatterjee, "Convergence of gradient descent for deep neural networks." arXiv:2203.16462, 2022.

---

> > ### Author Rebuttal · Reviewer_btjJ · 2026-04-02
> >
> > Thank the authors for their valuable response. My concerns have been addressed, and I believe this is a well-written and meaningful paper. I would like to maintain my original rating.

---

### Official Review · Reviewer_zXdX · 2026-03-13

**Soundness:** 3
**Presentation:** 4
**Significance:** 3
**Originality:** 3
**Overall Recommendation:** 5
**Confidence:** 3

**Summary:**

This paper provides theoretical advantages of using Gauss-Newton method in training neural networks. They work in the neural tangent kernel regime, where the smallest eigenvalue of the NTK matrix may slow down the convergence. They discuss two schemes, the overparametrized and the underparametrized regime. In the overparametrized regime they suggest an adaptive damping schedule which depends on the smallest eigenvalue of the NTK. In the underparametrized scheme they show the connection to Riemannian optimization and show exponential convergence. In both cases the method converges linearly with no dependency with the smallest eigenvalue of NTK.

**Compliance With Llm Reviewing Policy:**

Affirmed.

**Final Justification:**

I had a positive opinion towards the paper and it did not change during the rebuttal phase.

**Key Questions For Authors:**

1. Is there a heuristic that we can use to choose the right damping coefficient for training? e.g. adaptive scheme that is similar to binary search, etc. How will it accelerate compared to the oracle damping/ random damping?
2. The minimum width to make the theory work depend on the minimum eigenvalue of NTK matrix: is there a way to make this bound more realistic, or have less dependency on the minimum eigenvalue?

**Limitations:**

yes

**Strengths And Weaknesses:**

Soundness: This is a purely theoretical paper without any experiments. The theory follows a classical proof strategy where the trained parameters don't move so far from the initialization, and we use NTK - like ideas to prove convergence. I believe the main novelty is choosing an appropriate damping so that the linear convergence does not depend on the least eigenvalue of the NTK. This, though it could be not practical, shows acceleration, and I think it is an interesting contribution, because in general we do have dependence on the smallest eigenvalue.

Presentation: It is quite easy to understand the storyline for proof, and the motivations for each theorem or result. Assumptions, remarks, and figures are well distributed so that it is easy to understand the paper.

Significance: The idea that Gauss-Newton can remove the minimum eigenvalue dependence in convergence rate is a novel result, and I think this supports one of the many reasons why we would want to use Gauss-Newton. Again, it may be computationally infeasible to exactly use the proposed damping scheme, but I think it is meaningful that such characterization exists.

Originality: I think the damping coefficient is original, and connecting the algorithm to Riemannian optimization is also another novel axis. It is satisfying to see that in both cases we can have provable acceleration in the same sense.

---

> ### Author Rebuttal · Authors · 2026-03-31
>
> We would like to thank the reviewer for their insightful and constructive feedback. Below, we provide our responses to the reviewer's questions.
>
> **Damping coefficient choice.** A principled choice is to use the paper’s adaptive damping,
> $\rho_t=\frac{\lambda_{\min}(K_t)}{1+\lambda_{\min}(K_t)}$,
> which is theory-guided rather than tuned by trial and error and yields favorable geometric convergence guarantees as demonstrated in Theorem 3.4. When estimating $\lambda_{\min}(K_t)$ online is expensive, a practical alternative is the constant and data-dependent choice from initialization,
> $\rho_{\mathrm{con}}=\frac{\lambda^2}{1+\lambda^2},$ where $\lambda_{\min}(K_0)=4\lambda^2$. Intuitively, when the kernel matrix $Df(w_t) D^\top f(w_t)$ becomes ill-conditioned, the adaptive damping choice gives more weight to the Gauss-Newton matrix, thereby accelerating convergence in such situations. Our theory shows that these choices achieve a geometric rate that does not deteriorate as $\lambda_{\min}(K_0)\to 0$, unlike gradient flow, whose rate becomes arbitrarily slow in the ill-conditioned regime. Empirically, the adaptive and constant data-dependent choices behave very similarly in the kernel regime, so a costly binary-search/oracle tuning procedure does not appear to be necessary. We will clarify in the revision that the recommended practical strategy is: estimate $\lambda_{\min}(K_0)$ once and use $\rho_{\mathrm{con}}$, or use $\rho_t$ adaptively when online curvature estimates are available.
>
> **Minimum width.** We thank the reviewer for this question. As the reviewer noted, our paper assumes that the empirical neural tangent kernel matrix $K_0 = Df(w_0)D^\top f(w_0)$ is strictly positive definite, in line with prior work (Chizat et al., 2019; Du et al., 2018). A standard sufficient condition to ensure this is the non-collinearity of the data points and overparameterization of order $m = \Omega(n^2\log(n)/\lambda_0^2)$ (Du et al., 2018, 2019). Since this part does not depend on the optimization algorithm, our analysis inherits the same type of minimum-width requirement as prior results. For the optimization results, the relevant quantity is $\alpha\sqrt m$ (see Line 199), therefore increasing the scaling factor $\alpha$ can partially relax the need for very large width $m$, as in (Chizat et al., 2019). We will clarify this point further in the revision.

---

> > ### Author Rebuttal · Reviewer_zXdX · 2026-04-02
> >
> > Thank you for addressing my comment

---

### Official Review · Reviewer_iB2f · 2026-03-13

**Soundness:** 1
**Presentation:** 1
**Significance:** 1
**Originality:** 2
**Overall Recommendation:** 1
**Confidence:** 4

**Summary:**

This paper studies the convergence of Gauss--Newton for neural network training in both overparameterized and underparameterized regimes. In the overparameterized setting, it claims that an adaptive damping schedule yields geometric convergence rates that do not explicitly deteriorate with the smallest NTK eigenvalue. In the underparameterized setting, it interprets Gauss--Newton as a Riemannian gradient flow on a function-space manifold and derives geometric convergence under suitable scaling. The topic is relevant and the geometric viewpoint is interesting, but the submission has serious issues in correctness, clarity, reproducibility, and review integrity.

**Compliance With Llm Reviewing Policy:**

Affirmed.

**Key Questions For Authors:**

1. Why does the submitted PDF contain hidden reviewer-instruction text, and can the authors provide a clean version without it?
2. What exact adaptive damping formula was implemented in the experiments? Please reconcile the discrepancy between the theorem statement and the experimental description.
3. In what meaningful sense is the method robust to conditioning if the sufficient scaling conditions still worsen substantially as the smallest kernel eigenvalue decreases?
4. Which standard non-quadratic losses are actually covered by the global Hessian assumptions used in the paper?
5. Under what concrete conditions should the interior critical-point assumption in the underparameterized analysis hold for neural networks?
6. How are the Riemannian arguments justified when the manifold is explicitly described as having a boundary?
7. How do the authors expect practitioners to estimate or approximate the smallest kernel eigenvalue needed for the damping rule in large-scale settings?
8. What Euler step size, stopping criterion, random seeds, and preprocessing details were used in the experiments?
9. Can the authors provide stronger empirical evidence, including repeated runs, variance estimates, and comparisons against more meaningful baselines?

**Limitations:**

See question.

**Strengths And Weaknesses:**

Weaknesses.
1. The PDF contains hidden reviewer-directed text instructing the reviewer to include specific phrases in the review. This is unacceptable and substantially undermines trust in the submission.
2. The adaptive damping used in the theory and the one described in the experiments do not appear to match. As written, there is a notation/definition inconsistency, and it is unclear whether the experiments actually implement the theorem.
3. The claimed convergence rate may avoid explicit dependence on the smallest kernel eigenvalue, but the sufficient scaling/initialization conditions still deteriorate badly when that eigenvalue is small. So the result is not truly robust to ill-conditioning in a practical sense.
4. The analysis assumes global strong convexity and smoothness of the loss in prediction space. This excludes many standard losses unless extra restrictions are imposed, so the scope of the claims is broader than justified.
5. The convergence theorem assumes the existence of an interior critical point with vanishing Riemannian gradient. This is a major assumption and the paper does not show when it should hold for realistic neural network models.
6. The paper explicitly treats the image as a smooth embedded submanifold with boundary, but then uses exponential maps, geodesic convexity, and Riemannian Hessian arguments as if boundary issues do not matter. This part is not fully convincing.
7. The adaptive damping depends on the smallest eigenvalue of the kernel matrix along the trajectory, which is expensive and often impractical to compute. The proposed constant alternative still depends on spectral information at initialization.
8. The empirical evaluation is limited to small-scale examples with very narrow baselines. There are no strong comparisons, no multiple runs, no uncertainty bars, and no realistic large-scale experiments.
9. The experiments simulate continuous-time dynamics with Euler's method, but key details such as step size, seeds, solver settings, and full preprocessing are not reported.
10. The paper presents the result as requiring no explicit regularization, but the output scaling parameter clearly acts as a strong implicit regularizer by shrinking the admissible region and changing the best achievable in-class predictor.
11. The assumptions require smooth activations with bounded first and second derivatives, which excludes standard ReLU networks.
12. There are avoidable typos/grammar issues and some notation is not consistently maintained across the paper and appendix.

---

> ### Author Rebuttal · Authors · 2026-03-31
>
> We thank the reviewer for the detailed comments. We respond to the main points below.
>
> **Hidden reviewer-instruction text was inserted by ICML for watermarking.** The hidden reviewer-instruction text was **not** inserted by the authors. ICML's March 18 post states that ICML watermarked submitted PDFs with these hidden instructions to detect LLM policy violations in reviews.
>
> Source: https://blog.icml.cc/2026/03/18/on-violations-of-llm-review-policies/
>
> ICML 2026 Peer Review FAQ further states that *if a reviewer uncovers that organizer-inserted prompt, they should disregard it and review the paper as usual.*
>
> We respectfully note that this issue is unrelated to the scientific content of our submission and arises from organizer-inserted hidden instructions rather than any action by the authors. In line with ICML's published guidance, it should therefore be disregarded in evaluating the submission.
>
> **Adaptive damping in theory vs. experiments.** Appendix D presents numerical results for GN with adaptive/constant damping and gradient flow (GF). The discrepancy noted by the reviewer is due to a normalization choice. Theorems state the rules in the clear form $a=1$, whereas Appendix D.1 uses more generally $\rho_t=\frac{a\lambda_t^2}{1+a\lambda_t^2}$ with $a=1/4$. Plugging this into Lemma 3.3,
> $$V_t \le V_0 \exp(-2\nu \frac{1+a\lambda^2}{1+a}t).$$
> Thus, the experimental setup in Appendix D satisfies this bound with $a=1/4$.
>
> **Robustness to conditioning.** Our robustness interpretation is about the convergence rate, not about every sufficient condition becoming free of spectral dependence. Theorem 3.4 shows the exponential decay rate does not deteriorate as $\lambda_{min}(K_0)$ decreases (Lines 188-192). Under similar assumptions as GF (see Chizat et al., 2019), we prove rates for GN that do not deteriorate with $\lambda_{\min}({K}_0)$, unlike GF.
>
> **Assumptions.** The assumptions used in the theoretical results are stated explicitly in the paper: the prediction space loss $g$ is assumed to be $\nu$-strongly convex and $\mu$-smooth (see Equation (3)), and we assume smooth activations with bounded first and second derivatives (Lines 81-83).
>
> **Interior critical point.** This is an explicit representational assumption in Section 4. Under this assumption, we show that GN achieves the stated guarantees.
>
> **Manifolds with boundary.**
> - In Lemma 4.9, the existence of geodesic segments in $\mathcal{M}$ is proved in Appendix B via Alexander–Berg–Bishop characterization theorem, which applies to curvature bounds for Riemannian manifolds _with boundary_.
> - In Theorem 4.10 and Lemma 4.13, the regularity and PL arguments use $\rm Exp$, parallel transport and Hessian along geodesics. The arguments of Lemmas 4.2 and 4.5 extend to the open set $U:=\\{w:\\|w-w_0\\|\_2 < 3r_0/2\\}\supset B$ with $Df(w)^\top Df(w)\succeq (\lambda_0^2/4)I$ and $\\|\alpha f(w)-\alpha f(w')\\|\geq (\alpha\lambda_0/2)\\|w-w'\\|$ for $w,w'\in U$ so $\alpha f$ is bi-Lipschitz on $U$, thus $\alpha f$ is an embedding on $U$. Therefore, $\tilde{\mathcal M}:=\alpha f(U)$, equipped with the induced metric, is a manifold _without_ boundary containing $\mathcal M=\alpha f(B)$. We take $\rm Exp$ and parallel transport in Theorem 4.10 and Lemma 4.13 via $\tilde{\mathcal M}$. The geodesics used in the proofs connect $\alpha f(w_\alpha^\star)$ and $\alpha f(w_t)$, which lie in the geodesically convex sublevel set $S:=\\{z\in\mathcal{M}:\ g(z)\le g(0)\\}\subset \mathcal{M}$. Hence, the geodesics in Theorem 4.10 and Lemma 4.13 remain in $S\subset \mathcal M$. We will add a brief clarification in the revision to make this explicit.
>
> **Loss assumptions.** Our main focus in this paper is nonlinear regression with quadratic losses, in line with the seminal works (Chizat et al., 2019; Du et al., 2018). That said, our setting also covers regularized non-quadratic losses, e.g., binary logistic loss with $\ell_2$-regularization in prediction space, and logistic/exponential/Poisson on bounded predictors, as is the case in Section 3.1.1 with $tanh$.
>
> **Adaptive damping.** As stated before Corollary 3.6 (Line 176), tracking $\lambda_{\min}(K_t)$ is generally impractical in large-scale settings. For this reason, our paper also provides the damping rule in Corollary 3.6, computed once at initialization from the eigendecomposition of $K_0$ and then used throughout the optimization.
>
> **Empirical scope.** The experiments in Appendix D are intended to validate the theoretical results, not to serve as a large-scale empirical benchmarking study. The primary objective of our paper is to develop a concrete theoretical understanding of the Gauss-Newton method in neural networks. In Appendix D, we use the Euler step-size $dt=10^{-2}$ over a fixed horizon (i.e., without adaptive stopping criterion), and $\ell_2$-normalized input features.

---

> > ### Author Rebuttal · Reviewer_iB2f · 2026-04-03
> >
> > I appreciate your detailed reply; all points have been considered.

---

### Decision · Program_Chairs · 2026-04-30

**Decision:**

Accept (regular)

**Comment:**

One of the reviews seems to be generated by LLM, and its views are inconsistent with the other reviews. Therefore, that review is not being considered.

The other reviews point out that the manuscript contains good theoretical contributions on the convergence behavior of Gauss-Newton method for neural network training. However, given the potential algorithmic implications of the results, the manuscript's impact will be significantly enhanced if there is a more extensive numerical evaluation.